# Recent Excellent Optoelectronic Applications Based on Two-Dimensional WS_2_ Nanomaterials: A Review

**DOI:** 10.3390/molecules29143341

**Published:** 2024-07-16

**Authors:** Changxing Li, Dandan Sang, Shunhao Ge, Liangrui Zou, Qinglin Wang

**Affiliations:** Shandong Key Laboratory of Optical Communication Science and Technology, School of Physics Science and Information Technology, Liaocheng University, Liaocheng 252000, China

**Keywords:** WS_2_, optoelectronics, devices, nano, materials

## Abstract

Tungsten disulfide (WS_2_) is a promising material with excellent electrical, magnetic, optical, and mechanical properties. It is regarded as a key candidate for the development of optoelectronic devices due to its high carrier mobility, high absorption coefficient, large exciton binding energy, polarized light emission, high surface-to-volume ratio, and tunable band gap. These properties contribute to its excellent photoluminescence and high anisotropy. These characteristics render WS2 an advantageous material for applications in light-emitting devices, memristors, and numerous other devices. This article primarily reviews the most recent advancements in the field of optoelectronic devices based on two-dimensional (2D) nano-WS_2_. A variety of advanced devices have been considered, including light-emitting diodes (LEDs), sensors, field-effect transistors (FETs), photodetectors, field emission devices, and non-volatile memory. This review provides a guide for improving the application of 2D WS_2_ through improved methods, such as introducing defects and doping processes. Moreover, it is of great significance for the development of transition-metal oxides in optoelectronic applications.

## 1. Introduction

In comparison to their bulk counterparts, two-dimensional transition-metal disulfide compounds (TMDs) exhibit distinctive physical and chemical properties. Tungsten disulfide (WS_2_) is a highly significant member of the two-dimensional (2D) transition-metal dichalcogenides (TMDs) family and is regarded as an optimal candidate for optoelectronic devices due to its layer-related bandgap [1], high aspect ratio [2], high carrier mobility [3], excellent thermal stability [4], and excellent chemical stability [5]. WS_2_ is composed of a fundamental unit layer comprising a layer of W atoms sandwiched between two layers of S atoms. Each atomic layer is composed of atoms arranged in a hexagonal pattern, and W atoms combine with surrounding S atoms through strong ionic covalent bonds to form a relatively stable crystal structure. The most common crystal structures can be divided into two categories, depending on the stacking method. The metastable metal phase 1T-WS_2_ (octahedron) and the stable semiconductor phase 2H-WS_2_ (triangular prism) are the most prevalent. The second layer of S atoms in the 2H-WS_2_ lattice is located above the first layer of W atoms and exhibits spatial inversion symmetry (D^4^_6h_) [6]. The bandgap of WS_2_ exhibits a notable variation, ranging from a direct bandgap in the monolayer (2.1 eV) to an indirect bandgap in the bulk form (1.3 eV). The energy band exhibits significant valence-band splitting at the K point due to spin-orbit coupling, corresponding to direct transitions of two well-known A excitons (1.95 eV) and B excitons (2.36 eV) [7]. This results in strong photoluminescence and a large light absorption coefficient in a few layers. This material is suitable for the fabrication of various nanoelectronic and optoelectronic devices [8]. In contrast, the 2D WS_2_ surface lacks dangling bonds, which allows for the formation of advanced van der Waals (vdW) heterojunctions with other semiconductor materials [9]. The vdW heterojunction exhibits a clean, atomic-smooth, and defect-free surface without distortion or defects caused by a planar mismatch or thermal expansion [10]. In recent years, WS_2_ has been combined with a plethora of materials, including graphene, MoS_2_, perovskite, WO_3_, WSe_2_, ZnO, and Ti_3_C_2_T_x_, to form heterojunctions. These heterojunctions have been utilized to fabricate a diverse array of high-performance sensors, photodetectors, light-emitting diodes, and other devices. For instance, the combination of WS_2_ and ZnO has been demonstrated to enhance double exciton emission [11], while the combination of WS_2_ and WO_3_ has been shown to exhibit Warburg diffusion [12]. Furthermore, the combination of WS_2_ and Ti_3_C_2_T_x_ has been found to improve photodetection performance [13], and the combination of WS_2_ and diamond has been shown to increase field emission current density [14]. Furthermore, a series of techniques have been proposed to enhance the functionality of WS_2_-based devices, including ion-beam irradiation, surface modification, and local doping. For instance, the selective doping of electrons and holes can reduce dark current [15]. Biaxial strain WS_2_ is suitable for variable-frequency photoluminescence [16]. Furthermore, non-metallic dopants can reduce the potential barrier for electron tunneling [17]. In recent years, devices based on WS_2_ have achieved notable success in the field of optoelectronics. However, in recent years, researchers have conducted extensive reviews and discussions on TMDs, with a focus on the synthesis, properties, and applications of 2D WS_2_ [6]. To date, there is a paucity of comprehensive reviews that focus specifically on the application of 2D WS_2_-based optoelectronic devices. This review will primarily focus on the research on WS_2_ in various optoelectronic application devices and fields, including light-emitting diodes (LEDs), sensors, field-effect transistors (FETs), photodetectors, field emission devices, and storage devices. In light of the current state of progress, the challenges and prospective opportunities of 2D WS_2_ in diverse optoelectronic applications are elucidated. In light of the accelerated pace of recent discoveries, this review offers crucial insights for further advancing the high-performance development of new 2D WS_2_ optoelectronic devices toward practical applications in the near future.

## 2. Preparation of 2D WS_2_

The synthesis of WS_2_ can be broadly classified into two categories: bottom-up and top-down [18]. The top-down synthesis route encompasses several techniques, including mechanical exfoliation [19], ultrasound-assisted exfoliation [20], chemical exfoliation [21], and so forth. Among these methods, ultrasonic peeling is a process that utilizes the shear force generated by ultrasonic waves to separate the material into flakes [22]. The irregular nanosheet morphology is typically generated due to the instantaneous high pressure, which can cause damage to the material. Furthermore, ultrasonic exfoliation may result in the relative sliding of the WS_2_ atomic layer, which could potentially lead to a phase transition from 2H to 1T. Sharma et al. [23] observed this change during the ultrasonic exfoliation of WS_2_ quantum dots. In chemical exfoliation, lithium (Li) insertion results in the weakening of interlayer interactions, which are then dispersed in deionized water and exfoliated by ultrasound [24]. In contrast to MoS_2_, the lithiation process of WS_2_ is more challenging. Yang and Frindtin et al. [25] observed that the degree of lithiation only significantly increased when the temperature was above 50 °C. Subsequently, Tsai et al. [26] demonstrated that a phase transition from 2H to 1T occurred at a temperature of 350 °C.

Bottom-up methods include the hydrothermal method, sol–gel method, chemical vapor deposition, and pulsed laser deposition, among others [27]. For example, Liu et al. [28] generated a WO_3_ seed layer by pulsed laser deposition on a target substrate and then sulfided WO_3_ with thiourea at 250 °C to grow WS_2_ nanosheets. The synthesis of WS_2_ nanosheets by hydrothermal and sol–gel methods is challenging due to the tendency of WS_2_ to cluster into nanoparticles. In the process of improving hydrothermal methods, surfactants are often employed to form sheet-like micelles in the solvent, thereby facilitating the synthesis of WS_2_ nanosheets [29]. The hydrothermal method is another prevalent approach for obtaining the 1T metal phase WS_2_. In the experimental process designed by Ding et al. [30], a 1T/2H mixed phase dominated by 1t was formed. Chemical vapor deposition is a commonly used method to obtain high-quality TMDs. Zhang et al. [31] synthesized single-layer large single-crystal WS_2_ films through improved chemical vapor deposition. Additionally, chemical vapor deposition can be employed to directly evaporate WS_2_ to prepare nanostructures with varying morphologies and crystals [32].

## 3. Optoelectronic Applications Based on WS_2_ Devices

### 3.1. LEDs

WS_2_ compounds show a luminescence of about 2 eV at K points in a single layer, and its lack of suspended bonds, interlayer coupling, and inversion symmetry can produce strong PL and improve chemical stability [33,34]. Moreover, 2D monolayer WS_2_ typically has narrow linewidth exciton-absorption peaks situated at 450, 550, and 625 nm and a major exciton emission band located at 630 nm [35]. These excellent optical properties indicate that WS_2_ is an ideal candidate for LEDs.

#### 3.1.1. Polarized LEDs

WS_2_ has high exciton binding energy and valley polarization at the monolayer limit, which makes it widely used in strongly coupled exciton-polarization devices. Gu et al. [36] proposed a polariton LED at room temperature, based on a monolayer of WS_2_. As illustrated in Figure 1a,b, the device consists of a 12-period bottom-distributed Bragg reflector and a 40nm silver film, which are used as the bottom and top mirrors of the microcavity, respectively. In the active district, there is a tunnel region and two additional WS_2_ monolayers packaged with hexagonal boron nitride (h-BN). The tunnel region is composed of a vdW heterostructure, with a single layer of WS_2_ as a light emitter, a thin layer of h-BN on both sides of the single layer as a tunnel barrier, a single layer of graphene electrodes injected with electrons and holes, and a PMMA cover layer spin-coated on the vdW heterostructure. Figure 1c–f depicts the current-dependent polariton electroluminescence (EL) intensity. The total beam intensity increases with the increase in tunneling current. A weak electron emission from the polariton is discovered near the threshold bias (Figure 1d), whereas at an adequately high bias above the threshold, the polariton emission becomes noticeably brighter (Figure 1e). Figure 1c shows the polar coordinate diagram of EL intensity changing with angle, and the radiation patterns of the minimum driving current (green curve) and uppermost driving current (orange curve) remain almost unchanged. Figure 1f (black circle, left axis) presents the comprehensive strength under various drive tunnel currents, which demonstrates a nearly linear trend. It is possible that increasing the current to a sufficiently high value may result in the scattering of the polariton along the lower branch, which could in turn lead to a very restricted emission pattern due to polariton lasing. The external quantum efficiency (EQE) is also portrayed as a function of current density in Figure 1f (red circle, right axis). Overall, the device demonstrates the EL phenomenon of room-temperature polarized excitons and enables the injection and recombination of electrons and holes in WS_2_ monolayers (as light-emitting layers), providing a promising step towards the realization of ultra-fast microcavity LEDs.

The electrical control of the light helicity of LEDs is the basic operation of the light source in future light quantum technology. WS_2_ is prospectively a living material for chiral LEDs due to its valley contrast electronic system. Current WS_2_-based chiral LEDs are manufactured with magnetic materials/contacts, and their helicity is only regulated by magnetic fields at low temperatures of 10 K. Another method is to bring in a transistor structure to engender valley polarization EL in the P-N junction induced by an electric field. However, this type of device is still restricted to operating at low temperatures (usually lower than 80 K). Therefore, establishing a resultful method to control valley polarized EL at room temperature is a consequential challenge for subsequent device applications. Pu et al. [37] proposed a room-temperature chiral LED based on a strained-monolayer semiconductor. The helicity of the LED can be electro-regulated at room temperature, opening up a new way to construct a practical chiral light source based on a single-layer semiconductor for future quantum optical communication. The EL created by the hot carrier process and in the P-N heterojunction is located in the region adjacent to the electrical contact and at the edge of the heterojunction, respectively, and the spatial charge distribution generated by electrostatic doping is often uneven because of the existence of an electric field at the rim.

#### 3.1.2. Ordinary Monochrome LEDs

Expandability technology is a key step required for photodiodes to become practical. The preparation of TMD monolayers is one of the factors affecting their scalability. Andrzejewski et al. [38] proposed an extensible large-scale P-I-N LED based on WS_2_ monolayer metal–organic chemical vapor deposition (MOCVD). The full-color PL plot in Figure 2a shows the peak intensity in the 100 × 100 µm^2^ region. Figure 2b depicts the average PL spectrum at more than 10,000 measurement points, with samples exhibiting uniform PL emission. In Figure 2c,d, a histogram of full width at half maximum (FWHM) and peak wavelengths are given based on the assessment of 10,000 spectra. The narrow FWHM distribution of the spectrum is 12.6 ± 1.2 nm, corresponding to 42.7 ± 1.9 meV. In the red spectrum scope of 619.5 ± 0.9 nm, the light emission of the single-layer film is uniform and the wavelength changes are small. Figure 2e,f show the device composition and energy level chart, respectively. The PEDOT: PSS-covered indium tin oxide (ITO) can be regarded as a positive electrode or hole injection layer with a 5.2 eV work function, with single-layer WS_2_ acting as a luminescent layer, ZnO quantum dots acting as an electron injection layer, aluminum acting as a negative pole contact, and poly[*N*,*N*’-bis(butyl phenyl)-*N*,*N*-bis(phenyl)-benzidine] (poly-TPD) acting as an organic hole transmission layer. In addition, ZnO and poly-TPD can also act as electron and hole barrier layers to enhance the radiation recombination in the WS_2_ layer. When the bias voltage is 7 V, the current conduction of the LED is low, at 2.5 V, and the brightness is close to 1 cd/m^2^. Working in pulsating EL mode reduces dissipated heat and increases efficiency by nearly four times. In addition, it has the advantages of easy machining and scalability, paving the way for large-area 2D LEDs.

Using TMDs as semiconductor channels in Schottky diodes is often achieved using different chunks of metal as electrodes with diverse work functions, which results in thick electrodes, eliminating the concept of ultra-thin devices. Zhang et al. [39] propose a graphene bottom (GrB)–WS_2_-graphene top (GrT) heterostructure and provide the idea of obtaining a full 2D Schottky diode by preserving the ultra-thin device concept. Figure 3a,b demonstrates the I–V curve under various backdoor voltages, which shows obvious rectification behavior. Figure 3c plots the rectification ratio at different backdoor voltages, with a maximum of 100. The device (Figure 3d) exhibits N-type characteristics, and its threshold voltage is 10 V. The device has an asymmetrical Schottky barrier, which can be represented using the Schottky diode and series resistance model in Figure 3e. The rectification behavior exhibited by the device is explained well by Figure 3f. The bottom graphene is between WS_2_ and SiO_2_ and the top graphene is between WS_2_ and air, and the different dielectric environments of the two lead to differences in the doping level of graphene, which leads to differences in the Schottky barrier, and finally leads to the displayed rectification behavior. In addition, an uninterrupted bias voltage exerted on the device produces a mighty red-field EL in the WS_2_ region on both graphene electrodes, which can be used in photodiodes, as displayed in Figure 3g. The device produces a red EL bar, reposed between the two poles, whose homologous intensity distribution is displayed in Figure 3j in a bell-shaped profile, which can be treated as a Gaussian distribution. Figure 3h describes the time reliance of the current: first, the current passing through the device rises rapidly, and over time, it gradually decreases over the next 16 s, with an average current of 2.16 × 10^−5^ A over this period, indicating that material degradation occurs when carriers are laterally injected into the semiconductor WS_2_ exposed to air. This work helps provide the idea of obtaining a full 2D Schottky diode by retaining the ultra-thin device concept.

#### 3.1.3. Wavelength Adjustable LEDs

In alternating-current (AC) LEDs, the combination of electrons and holes can only occur at the contact edge between the electrode and the 2D single-layer material, so the effective EL area is small. Zhu et al. [40] developed a hybrid continuous-pulse injection method based on WS_2_ LEDs. EL emission was observed at 25 µm from the contact edge, significantly increasing the effective luminescence area. In addition, the wavelength-tunable AC-driven WS_2_ single-layer LED device is demonstrated. By controlling the frequency response of the drive signal, the wavelength of the WS_2_ single-layer LED device can be switched from exciton emission to three-ion emission and finally to defect emission, with tunable capability. Functional optoelectronic devices based on TMDs are closely associated with the band gap of the materials. The usual methods for adjusting the band gap of materials include gating, strain, and dielectric engineering; however, the control stability of these methods is profoundly restricted. Pu et al. [41] used a composition-gradient single-layer WS_2(1−x)_Se_2x_ alloy (x varying from 0 to 1) to fabricate an electrolyte-based continuously color-adjustable LED. This composition gradient directly causes the luminous energy of the same sample to vary between 2.1 and 1.7 eV, and this LED does not require any outer optical and mechanical parts to regulate the emission color, providing new insights to explore the application of broadband optoelectronic devices.

#### 3.1.4. Summary

Despite the progress made in LED devices, designing polarization for single-layer EL emission remains quite challenging. Due to the valley degree of freedom of the monolayer, it has been proven that circularly polarized LED devices based on P-I-N-structure mechanical exfoliation and chemical vapor deposition monolayers have been developed [41]. However, developing linearly polarized LEDs requires anisotropic materials—for example, recently achieved linearly polarized LEDs that rely on anisotropic black phosphorus (BP). However, their stability is poor. WS_2_ has high anisotropy and stability, and using WS_2_ instead of BP is expected to achieve high-performance linearly polarized LEDs [42]. Nanolight sources are an active research field, and graphene-based electrically driven thermal emitters exhibit poor performance in low-efficiency broadband thermal emission. WS_2_ can exhibit strong bound excitons and a short lifetime at room temperature, and single-layer-based LEDs are expected to achieve the ultra-fast modulation of emitted light [43]. The luminescence behavior of WS_2_ is closely related to its thickness. Single-layer WS_2_ exhibits strong luminescence due to radiative recombination at the K point due to valence-band splitting. The nonradiative recombination and radiative recombination of multi-layer WS_2_ often compete with each other, resulting in a decrease in luminescence efficiency [44]. In addition, introducing strain or defects into semiconductor materials can significantly adjust their bandgap, thereby affecting their electroluminescence phenomenon. In this manuscript, we present, in the form of a table, the synthesis methods and applications of representative WS_2_ with different morphologies for intuitive understanding. Table 1 is as follows:

### 3.2. Sensors

From monitoring air quality to testing food, gas sensors have become a ubiquitous aspect of modern life [45]. Because of its large surface area, multipurpose surface chemistry, and ability to accurately detect gases in the environment, the 2D WS_2_ nanostructure has attracted tremendous attention for its particular combination properties and comprehensive applications in gas sensors [46]. Furthermore, except for a layer-associated band gap of 1.3 to 2.05 eV, WS_2_ has a high electron mobility because of its low effective mass, and the layered organizations formed by WS_2_ can be readily peeled off into monolayers or several layers, which can strengthen its electrical properties [47].

#### 3.2.1. Gas Sensors

Ullah et al. [48] investigated the interaction of various analytes with diverse structures, bonding, and molecular sizes (toluene, acetone, ethanol, and water) with laser-stripped WS_2_ sensing materials. The results indicate that the response currents of the sensor are different for all analytes, the response currents of ethanol and water are particularly high, and the detection limits are also different. Although water and ethanol include the same O-H, there is a small difference between the supreme current values of water and ethanol, and the response speed is also different, due to the smaller molecular size of water compared to ethanol. These results can be understood by the occupation of interaction sites—larger-size analyte molecules take up more regions on the sensing medium and may include many interplay sites, and analytes with fewer molecules (ethanol and water) do not saturate the interplay sites as rapidly. This offers an overall picture of analyte–sensor interplay that could help advance semiconductor sensors.

Liu et al. [49] prepared WS_2_/WO_3_ heterojunction nanosheets using the in situ oxidation method, as shown in Figure 4. By adjusting the concentration of the oxidizer and oxidation time to adjust the ratio of WS_2_ and WO_3_, three different samples were obtained with the ratios 1:0.7, 1:1.6, and 1:2.5, which were named WS_2_/WO_3_-2, WS_2_/WO_3_-3, and WS_2_/WO_3_-4, respectively. The specific surface area of the heterojunction was determined by the adsorption–dissociation experiment to be about 32.76 m^2^/g, as shown in Figure 4e. Figure 4g–j presents the response curve of the sensor to 20 ppm acetone at dissimilar temperatures. At 100 °C, the sensor possesses a smaller response to acetone and a longer response/recovery time. When the temperature rises to 150 °C, the chemical activity of the material is increased due to the temperature, which makes the performance of the sensor significantly improved. As the temperature continues to rise to 200 and 250 °C, the sensing performance decreases significantly, which is due to the difficulty of oxygen adsorption and electron transfer on the surface of the material. In addition, there was no significant difference in the response of the sensor to acetone detection in five repeated adsorption and desorption cycles, indicating that the gas sensor had good repeatability, and the response of the sensor did not decrease significantly within 30 days of strong stability. It is suggested that using heterojunctions can be considered an advantageous method for improving the sensitivity of gas sensors.

Formaldehyde (HCHO) is a classical colorless nocuous gas with an irritating odor, causing huge danger to the central nervous system and immune system of the human body. Zhang et al. [47] prepared a room-temperature HCHO sensor based on a Ni-doped In_2_O_3_/WS_2_ hybrid material. The sensor has the uppermost response to HCHO. However, the response of the sensor decreases with the increase in humidity, as the H_2_O molecule reacts with the oxygen anion adsorbed on the surface of the compound to constitute the hydroxyl group (OH), leading to a decrease in the adsorption of the oxygen anions. In practical applications, humidity compensation may be required to achieve high-accuracy HCHO detection. The preparation of Ni-doped In_2_O_3_/WS_2_ nanomembranes offers a new route for the development of promising formaldehyde-sensing hybrid materials. Zhang et al. [50] prepared an ethanol sensor based on WS_2_/WO_3_ utilizing a layer-by-layer self-assembly method. To heighten the adhesion strength, two layers of Poly (diallyldimethylammonium chloride) /polystyrene sulfonate (PDDA/PSS) were deposited on polyethylene terephthalate (PET) substrate, and then the sensor was immersed in WO_3_ and WS_2_ solution as the sensing layer. In the gas sensing test, the sensing characteristics are optimal when the ratio of WS_2_ to WO_3_ is 1:1. The test was able to detect 1 ppb-50 ppm of ethanol gas with a ppb level detection limit. It also has better gas selectivity and response/recovery times than WS_2_ and WO_3_ gas sensors. This sensor can be used for the scene of ppb-grade ethanol gas.

Nitric oxide (NO) is the most universal atmospheric pollutant, mainly engendered by industrial emissions and vehicle exhaust. Lv et al. [51] prepared an NO gas sensor based on an SnO_2_/WS_2_ heterogeneous structure using the hydrothermal method. The sensor has an optimal working temperature of 75 °C (5 ppm NO), as shown in Figure 5. After five cycles, the percentage of change is below 2.7%, the response/recovery time has no significant variation, and the response time and recovery time are 96 and 17 s, respectively. These excellent gas-sensitive properties result from oxygen vacancies in the SnO_2_/WS_2_ hybrid, non-stoichiometric tin oxides (SnO_2−x_), and strong electron interactions. Since the work function, electron affinity and band gap of SnO_2_ and WS_2_ are different, when WS_2_ and SnO_2_ contact, electrons will flow from SnO_2_ to WS_2_ and finally reach Fermi-level equilibrium and produce a depletion layer. When the device is exposed to NO, a chemical reaction will occur on the surface of the material, and NO will take away electrons from the SnO_2_ layer to make the energy barrier higher, leading to an increase in the resistance of the SnO_2_/WS_2_ material. It is worth mentioning that the SnO_2_/WS_2_ hybrid material can better regulate carrier transfer under the effect of heterojunctions, so it can achieve a better sensor response.

Metal-oxide–semiconductor (MOS) gas sensors have drawn much attention because of their advantages of easy manufacturing, low cost, high sensitivity, and reliable stability. The comparatively low molecular activity on MOSs at room temperature tends to inhibit the response of the sensor. Xu et al. [52] proposed a NO_2_ gas sensor based on the WS_2_ nanosheet/carbon nanofibers (CNFs) complex structure. Because the WS_2_/CMFs composite structure provides more active sites for NO_2_, the gas sensor has a more sensitive response to NO_2_. The response at 10 ppm NO_2_ is 2.11, which is higher than that of the WS_2_ gas sensor and the CNF gas sensor. In addition, the detection of NO_2_ also has excellent selectivity and stability. This work supplies fresh clues for the exploitation of high-performance gas sensors by designing edge-rich TMD nanostructures. Moumen et al. [53] reported a room-temperature NO_2_ gas sensor with a P-P 2D layered structure synthesized from shed hexagonal WS_2_ and hexagonal tungsten selenide (WSe_2_). Sensors based on WS_2_-WSe_2_ showed long-term stability even at high humidity levels (up to 90%). This is due to the strong affinity between the P-P structure and NO_2_ gas. In addition, the P-P structure at the interface leads to increased carrier charge mobility and larger specific surface area with more active adsorption sites, so the response rate and response speed of the WS_2_-WSe_2_ sensor to NO_2_ are enhanced. These results suggest that WS_2_-WSe_2_ has the potential to monitor NO_2_ at room temperature with high selectivity.

Hydrogen sulfide (H_2_S) is a highly toxic and foul-smelling gas that is also a naturally occurring waste in crude oil, sewers, waste treatment industries, geothermal power plants, and food. Li et al. [54] implemented the H_2_S gas sensor by decorating the CNF template CuO with WS_2_ nanosheets (W-Cu-C), using it as the sensing layer of a chemo-resistive microelectromechanical system sensor. Compared to the original CuO at 160.5 °C, the designed W-Cu-C sensor is 37 times more responsive to 0.5 ppm H_2_S at a lower operating temperature (100.1 °C). The response to 0.5 ppm H_2_S was a recovery speed of 37.2/33.9 s with a low detection limit (200 ppb) and super-low power consumption (8 mW). This provides a new way to achieve ultra-sensitive trace H_2_S detection. Sakhuja et al. [55] used a microwave irradiation-assisted solvothermal process to prepare H_2_S gas sensors based on sulfate/oxide heterostructures co-existing with WS_2_ and ZnO. The process is an effectual one-step route to achieve heterogeneous structures at low temperatures and low cost, with scalability and CMOS compatibility. The sensor is five times more responsive to H_2_S than the original WS_2_ and ZnO. This process is easy and rapid and requires a low thermal budget, making it tempting for industrial-scale production. Chen et al. [56] explored the sensing characteristics of a WS_2_-based gas sensor for four gases (SO_2_, SOF_2_, SO_2_F_2_ and H_2_S). The order of sensitivity is H_2_S, SO_2_, SO_2_F_2_, and SOF_2_ in descending order; this is because of the intense chemical interplay of SO_2_ and H_2_S with the edge structure, while the chemical interaction of adsorbed SOF_2_ and SO_2_F_2_ is weak, and the corresponding detection limits are calculated as 773 ppb, 1.30 ppm, 4.57 ppm, and 4.82 ppm. For the four gases, the linearity of WS_2_ is higher when the concentration is 0~50 ppm. However, once the concentration attains 100 or 150 ppm, the saturation phenomenon becomes observable. This study provides a microscopic mechanism for TMDs as sensing materials.

Ammonia (NH_3_) is a deleterious gas present in the atmosphere and water, and a low-cost, lightweight, simple-to-operate-and-move NH_3_ sensor has important applications. Sharma et al. [57] prepared different proportions of SnO_2_ quantum dot-decorated 2D WS_2_ nanosheets by liquid phase stripping technology and further prepared NH_3_ sensors based on these nanosheets. When the NH_3_ concentration is 10 ppm, the sensor response of the sample with the WS_2_:SnO_2_ ratio of 10:1 improves the most (~850%), which is 8.5 times that of the WS_2_ sensor. Singh et al. [58] designed and manufactured NH_3_ and NO dual gas sensors utilizing WS_2_/multi-walled carbon nanotubes (WS_2_/MWCNT) as channels (Figure 6). As shown in Figure 6j–m, the D2 (WS_2_/MWCNT (10 mg)) device exhibits delayed recovery for both NH_3_ and NO. D2 has the same resistance change as D1 (WS_2_/MWCNT (5 mg)). Figure 6n,o summarizes the selectivity for all analytes. On the whole, the composite can be applied for NH_3_ sensing as low as 0.1 ppm and NO sensing as low as 5 ppb and has excellent moisture resistance and selectivity, which has applications in environmental detection and medical diagnostics.

Ou et al. [59] synthesized mesoporous WS_2_/MoO_3_ hybrid NH_3_ sensors in an easy two-step hydrothermal way. Four sensors, named WM-5, WM-10, WM-20, and WM-30, were prepared by adding different amounts of WS_2_ (5, 10, 20, and 30 mg, respectively). Figure 7 shows the response of WS_2_, MoO_3_, and WS_2_/MoO_3_ to 3ppm H_2_S, C_3_H_6_O, NH_3_, SO_2_, CO_2_, and CO at room temperature. Compared with other gases, these sensors respond more significantly to NH_3_ with good gas selectivity. WS_2_/MoO_3_ exhibited similar P-type behavior and higher electrical conductivity compared to MoO_3_. WS_2_/MoO_3_ showed better reproducibility and a greater NH_3_ response, among which WM-10 had the highest response of about 31.58%, which was 17.7 times that of MoO_3_ and 57.4 times that of WS_2_ (Figure 7). Because of the dissimilarity in work function, when the two come into contact, electrons will be transferred from WS_2_ to MoO_3_ when exposed to NH_3_, NH_3_, and O^2−^ and undergo redox reactions, narrowing the depletion layer, as shown in Figure 7f. Overall, WM-10 has the highest transient response and fastest response time, outperforming single-component sensors (WS_2_ or MoO_3_). The heightened response of WS_2_/MoO_3_ sensors is because of the porous structure and abundant heterojunction. Additionally, WS_2_/MoO_3_ showed exceptional stability and selectivity to NH_3_.

#### 3.2.2. Flexible Sensors

Vehicle exhaust and exhaust gas from factories are the principal sources of air contamination, which seriously jeopardize human health, and the application of low-energy consumption sensors in life is crucial. Kim et al. [60] implemented a self-heating flexible gas sensor with low power consumption based on WS_2_-SnO_2_ core-shell composite (C-S) nanosheets (NSs) by atomic layer deposition (ALD). By comparing the response of different shell thicknesses to CO, they determined that the best shell thickness is 15 nm, and the maximum response is eight. The same method is used to determine the optimal shell thickness for NO_2_ detection, which is 30 nm. The sensor is sensitive to reducing gases such as CO, C_7_H_8_, HCHO, NO_2_, and SO_2_. In addition, as the core WS_2_ is deposited on the flexible substrate, the difference in response between the sensor bent 0 times and the sensor bent 1000 times is negligible, indicating that the sensor can also be applied in wearable equipment. Hao et al. [61] used sputtering technology to deposit a WS_2_ film covering a 3nm thick Pd layer on an ultra-thin Si substrate and prepared a bendable PD-WS_2_ /Si heterojunction H_2_ sensor. By comparing the response of different WS_2_ layers to H_2_, it was found that the response reaches the maximum when WS_2_ is five layers. Testing with different gases (N_2_, O_2_, H_2_, CO, NH_3_, H_2_O, and ethanol) and different humidities (25%RH and 89%RH) found that the sensor responds most strongly to H_2_, and the H_2_ response decreases slightly as the humidity level increases. This is attributed to the fact that the sensor surface is covered with H_2_O molecules, reducing the contact area between the H_2_ and the Pd. In addition, the sensor has no obvious influence on the sensing characteristics in the bending state, showing good stability. This work supplies an effective way to develop powerful, flexible, ambient H_2_ sensors.

The evolving Internet of Things demands lower power consumption for high-performance sensors and intelligent wireless sensors. Ma et al. [62] prepared a self-powered flexible NO_2_ gas sensor based on the heterogeneous structure of WS_2_/graphene by Ga^+^ ion irradiation-control defect engineering. The sensor possesses a detection limit of 50 ppb for NO_2_ and a response time of 110 s. In addition, the sensor can be driven under the photovoltaic effect and does not require an external power source. The sensing properties of the defect-engineered heterostructures remained stable even after 1000 bends, confirming their application potential as flexible devices. Zhang et al. [63] propose a wireless-powered, multifunctional, wearable humidity sensor based on a reduced graphene oxide (RGO) and WS_2_ heterojunction. The sensor is mainly composed of an LC circuit, a dielectric layer, and a sensing layer, which can transmit real-time wireless data. When the sensor is bent from 0° to 50°, the reflection value of the sensor diminishes sharply with the growth of the bending angle, but the resonant frequency of the sensitive parameter is slightly offset, which indicates that the sensor is apposite for wearable. When the relative humidity is below 35%RH, the sensitivity reaches approximately 11 kHz/%RH. When the humidity is 35%~65%RH, the sensitivity reaches approximately 140 kHz/%RH. When the humidity is greater than 65%RH, the sensitivity reaches approximately 120 kHz/%RH. The exceptional performance results from the significant electron transfer on the sensing material. This transfer creates a higher number of active sites available for water molecules to bind to. These results indicate that it has broad application potential as a wearable device.

#### 3.2.3. Summary

In comparison to metal-oxide resistance gas sensors, gas sensors based on WS_2_ are capable of operating at lower temperatures [64]. However, their optimal working temperature remains higher than room temperature. For instance, the optimal working temperature for acetone gas sensors based on WS_2_/WO_3_ is 150 °C, while for NO gas sensors based on SnO_2_/WS_2_, it is 75 °C [49,51]. Although the self-heating effect enables operation at the optimal temperature, the development of room-temperature gas sensors remains a necessity. It has been demonstrated that the doping and modification of noble metals can significantly enhance the responsiveness of sensing materials to target gases [65]. Furthermore, in humid environments, the presence of water molecules on the surface of sensors can diminish their sensing capabilities by altering the physical properties, competitive adsorption, and chemical adsorption of the sensing material [66]. The effect of humidity can be significantly mitigated by coating with hydrophobic coatings with high breathability. For instance, Gao et al. [67] employed polydimethylsiloxane (PMDS) to coat TiO_2_ nanotube arrays. The morphology of sensing materials is typically multi-mesoporous, nanoflower, or vertically grown heterojunctions, which can expose more active sites in limited space to enhance sensing performance. Furthermore, research has demonstrated that a reduction in the number of layers can facilitate a faster recovery time, while an increase in the number of layers can enhance the response rate [68]. Consequently, the performance of sensors is contingent upon their morphology and thickness. This manuscript presents the synthesis methods and applications of representative 2D WS_2_ with different morphologies in the form of a table, which facilitates intuitive understanding. The Table 2 is presented below.

### 3.3. Transistors

A transistor has the functions of amplifying, rectifying, modulating signals, etc., and plays an extremely essential role in the field of electrical appliances. For high-performance transistors, it is crucial to have channels with thin widths to guarantee efficient gate control and gate length scaling [69]. WS_2_ is a promising channel material due to its lower effective mass and higher mobility that allows transistors to be further miniaturized due to its atomically thin nature [70,71].

#### 3.3.1. Ordinary Metal-Oxide–Semiconductor FETs

Single-layer TMDs are inherently constrained by quantum constraints and low state densities, resulting in a high Schottky barrier height and contact resistance and low carrier mobility, limiting the achievement of high-performance FETs. Li et al. [72] reported an effective method for implementing high-performance atomic-thin FETs by CVD 3R stacked dual-layer WS_2_. The effective mobility of 3R dual-layer WS_2_ exceeds that of 2H dual-layer WS_2_ by 65%. Figure 8a is a schematic of a 3R-WS_2_ short-channel FET in which 3R-WS_2_ is used to construct a channel about 1 micron in width. Figure 8b describes a high-resolution transmission electron microscope (HRTEM) cross-section image of the device with a channel length of about 50 nm. Figure 8c shows the transmission characteristics of the device at 300 and 4.3 K, and it can be seen that it has an on/off ratio of 10^8^ when the V_ds_ is 1V. Figure 8d shows the output characteristics of the equipment at 300 and 4.3 K. The results show that the maximum I_ds_ of the device can reach 480 μA/μm under the condition of V_ds_ = 1 V, showing good saturation performance. At the same time, the device obtains an ultra-low conduction resistance (R_on_) of 1 kilohm·μm at 300 K. In Figure 8e,f, R_on_ and the on-state current (I_on_) of WS_2_ transistors in the other literature and 3R-WS_2_ transistors are compared. This study shows that double-layer TMDs with 3R stacks offer a promising path to fulfill high-performance nanoscale transistors.

Recent transport simulations show that double-layer TMDs have better device performance than single-layer TMDs due to their higher inherent mobility and state density. In addition, the double-layer material is more immune to manufacturing-induced damage or interface scattering, resulting in a smaller effective Schottky barrier height (SBH) and reduced Fermi-level pinning. Shi et al. [83] fabricated high-performance FETs based on single- and double-layer WS_2_ by CVD. The average field-effect mobility of the double-layer WS_2_ FET is twice that of the single-layer FET. In addition, without doping, the double-layer WS_2_ short-channel transistor has high conducting current and ultra-low contact resistance at a channel length of 18 nm and V_ds_ =1 V, reaching 635 μA/μm and an Rc as low as 0.38 kΩ·μm. It also reached a record saturation speed of ~3.8 × 10^6^ cm/s at room temperature. The intrinsic robustness of double-layer WS_2_ crystals is revealed, which has good potential for integration with traditional manufacturing processes.

Realizing the full potential of WS_2_ requires an understanding of the high-field transport in short-channel devices, where heat removal is critical. Shi et al. [84] demonstrated a high-performance N-type single-layer WS_2_ FET with high-quality BeO dielectrics. A schematic of the transistor is shown in Figure 9a, using HfO_2_ or BeO as the gate and WS_2_ as the semiconductor channel. Figure 9b shows the transfer characteristics of HfO_2_ and BeO dielectric single-layer WS_2_ devices with a channel length (L_ch_) of 1 μm, both of which exhibit similar N-type input performance with an on/off ratio of up to 10^8^. Figure 9c shows the output characteristics of a device with a channel length of 1 μm. The drain current of a WS_2_ transistor based on BeO media is 65% of that based on HfO_2_ media at the same carrier density. The output characteristics of the 50nm channel length device are shown in Figure 9d. The device with HfO_2_ as the gate shows obvious negative differential resistance behavior in the output characteristics due to the self-heating effect. In contrast, the devices with BeO as the gate do not observe negative differential resistance behavior but exhibit good drain saturation and a record-high room-temperature current density (>400 μA/μm), which is caused by the high thermal conductivity of BeO. Figure 9e shows a comparison of the statistical analysis of the mobility of the WS_2_ devices of HfO_2_ and BeO. The average field-effect mobility of HfO_2_-based devices is 16 cm^2^/V·s, and that of BeO-based devices is 26 cm^2^/V·s. Contact resistance (R_c_) is a key performance indicator of a transistor, as shown in Figure 9f. Because the BeO-based device has a higher amount of migration, its R_c_ is smaller than that of the HfO_2_-based device—about 1.1 kΩ·μm. This study shows that the use of high thermal gate media is propitious for improving device performance.

Hou et al. [85] reported the electrical property modification of WS_2_ monolayer materials by low-energy Ar^+^ plasma processing. By comparing the WS_2_ FETs with the top electrode and bottom electrode, it is found that after low-power (1W) Ar^+^ plasma processing for a short time (≤16 s), surface contamination is eliminated and carrier SBH and effective mass are reduced, thereby enhancing mobility and reducing on-voltage. This work provides an efficient tactic for controlling the Schottky barrier and mobility of single-layer WS_2_ FETs.

To enhance the performance of WS_2_ FETs, Liu et al. [86] introduced two Cu doping strategies and revealed the influence of Cu doping on the carrier transport of WS_2_ at the semiconductor–metal interface. Cu charge transfer can markedly alter the Fermi level of WS_2_ along the channel and greatly reduce the work function of Cu at the semiconductor–metal interface with the decrease in the contact barrier. The Schottky–Ohm contact transition can strengthen carrier injection at the semiconductor–metal interface and lead to a sharp fall in contact resistance, increasing in electron mobility. Compared with the original WS_2_ FETs, the contact resistance is lessened by one to three orders of magnitude, so both generalized Cu atom doping and local Cu decoration can realize the Schottky-to-ohmic contact transition, thereby increasing the electron mobility by five to seven times. This proves that the introduction of transition metals is an effective technique to improve the carrier transport of FETs.

The structural defect of SV in TMDs is the internal factor for the significant reduction of carrier injection and transport, resulting in significantly lower mobility than its intrinsic mobility. Jiang et al. [87] reported on a method for the performance enhancement of WS_2_ FETs by mending intrinsic sulfur vacancies. In the nitrogen plasma treatment, due to the stability of adsorption, nitrogen atoms tend to fill SV and compose W-N bonds. In addition to eliminating defects in WS_2_ materials, the nitrogen plasma process enhances the surface characteristics of the material. The FET based on nitrogen plasma machining has an ultra-high field-effect mobility (184.2 cm^2^V^−1^s^−1^) and a low threshold voltage (3.8 V). The study provides an approach to improving TMD electronic device performance in practical electronic applications.

#### 3.3.2. Metal–Insulator–Semiconductor Contact FETs

Zheng et al. [88] report a resultful method for reducing the SBH of WS_2_ FETs source and drain (S/D) contacts using an ultra-thin Al_2_O_3_ interface layer between metal and WS_2_. The results show that inserting ultra-thin Al_2_O_3_ between the metal and WS_2_ can move the Fermi level and reduce the SBH at the interface. When the thickness of Al_2_O_3_ is 1.1 nm, the FET has the largest current passing performance, increasing the switching ratio and enhancing the field-effect mobility, and the conduction current of the FET with 1.1 nm Al_2_O_3_ is about 50 times higher than that without adding Al_2_O_3_, paving the way for the easy integration of TMD electrons and optoelectronics with current semiconductor technology.

Phan et al. [89] studied FETs with direct metal–semiconductor (MS) and metal–insulator-semiconductor (MIS) contact structures. Figure 10a shows the schematic diagram and SEM image of the MIS-structured FETs in which a few layers of WS_2_ are used to construct the channel. Figure 10b, c is the Raman spectra of WS_2_ and h-BN, while Figure 10d,e is the AFM images of WS_2_ and h-BN. Combined with Figure 10b–e, it can be seen that the number of layers of WS_2_ is about 4 layers, and h-BN is about 1 layer. Figure 10f–i shows the current-voltage characteristics of different contact devices with chromium (Cr) and indium (In) as electrodes and h-BN as intercalations. In the output characteristics of indium electrode contacts, MS contacts exhibit ohmic behavior, whereas MIS contacts exhibit nonlinear behavior due to the presence of back-to-back Schottky diode structures. In contrast, when Cr is deposited as an electrode, MS contacts behave as Schottky contacts, while MIS contacts behave as ohmic contacts. MIS contacts with Cr electrodes demonstrate superior performance in comparison to MS contacts within electrodes and MS contacts with Cr electrodes. By further investigating the effect of h-BN of different thicknesses on device performance, it was found that the contact resistance of MIS contacts was significantly reduced by about 10 times compared to MS contacts. At 300 K, the electron mobility of single-layer h-BN reaches about 115 cm^2^V^−1^s^−1^, about 10 times higher than the MS contact. This work presents an efficacious methodology for the reduction of contact resistance.

#### 3.3.3. Dual Door Control FETs

Thus far, multiple researchers have published findings regarding devices composed of TMDs, featuring proportional channel lengths. While some have demonstrated satisfactory cutoff state behavior, the performance of the conduction state has yet to meet the predicted level. For this reason, Pang et al. [82] reported that WS_2_ FETs fabricated on exfoliated multistory channels with outstanding on-state and off-state performance. Achieved record high conduction current (I_ON_) and ultra-low contact resistance (R_C_) at scaled overdrive voltages (V_OV_ = V_GS_ − V_TH_), R_C_ reached ~500 (Ω×μm) and I_ON_ greater than 600 (μA/μm) at V_DS_ = 1 V and V_OV_ = 2 V. In addition, the data statistics of FETs with different channel lengths show that they show good off-state, small threshold voltage (V_TH_) changes, close to desirable subthreshold slope, and small drain-induced barrier reduction. Finally, different channel thicknesses within the range of 2.1 nm to 7 nm were thoroughly evaluated regarding their short-channel effects and on-state currents. It was found that a WS_2_ body thickness of 2.1 nm (three layers) exhibited superior performance in both on-state and off-state.

Jin et al. [77] fabricated WS_2_ metal-oxide–semiconductor FETs with semi-metallic Bi contacts with a saturated drain current of up to 245 μA/μm (340 μA/μm) at 78 K, with an S/D spacing of 0.32 μm and an on/off ratio up to 10^10^. This is a powerful demonstration of the latent energy of WS_2_ based on CVD in achieving high-performance devices.

#### 3.3.4. WS_2_ P-Type Channel FETs

Acar et al. [90] used the single-step magnetron sputtering method to prepare WS_2_ films of different thicknesses by changing the sputtering time (1 s, 5 s, 10 s, and 30 s) and studied FETs based on these WS_2_ films. All the prepared film samples were single-phase WS_2_. The S/W atomic ratio was about 1.15–1.30, showing a sulfur deficiency phenomenon. The FET device prepared for 1 s exhibits P-type channel behavior, pico-ampere-level turn-off currents and an on/off ratio spanning four orders of magnitude. The FET devices prepared for 5 s, 10 s, and 30 s achieve bipolar behavior and achieve a mobility of about 10–20 cm^2^/ (V s). The sputtered WS_2_ material grown in a single step is beneficial to the application of FET devices and simplifies the fabrication of devices.

Many efforts have been made to fabricate FETs using single-layer WS_2_, but the devices produced often present low mobility and N-type conductive behavior, hindering complementary metal-oxide–semiconductor (CMOS) application and integration. Xie et al. [91] proposed the VLS growth of WS_2_ with adjustable morphology for P-type single-layer FETs. WS_2_ with different morphologies can be readily synthesized by adjusting the temperature, sulfur introduction time, and growth time. Transistors based on this growth mode exhibit unique P-type behavior due to the replacement of W atoms with Na atoms, which generates a defective state because Na atoms cannot provide enough electrons to saturate S, resulting in the emergence of extra holes in WS_2_, ultimately leading to the P-type conductivity of WS_2_. This work paves the way for the integration of P-type WS_2_ into large-scale CMOS and other wearable devices.

Liang et al. [92] studied the electron transport characteristics of SiO_2_/WS_2_ FETs regulated by electron-beam irradiation at gate voltage (V_gs_) and proposed a possible mechanism. The change of transport characteristics induced by irradiation depends largely on the V_gs_ during irradiation. High positive V_gs_ significantly increase the electron doping, while negative V_gs_ significantly decrease the doping level. A good deal of electron-hole pairs were excited by the WS_2_ wafer and dielectric layer during irradiation. Under positive V_gs_ irradiation, the resulting electrons may spread through the dielectric layer under the V_gs_, while the holes may be seized by defects in the dielectric layer and there are no other electrons or molecules to neutralize them. Therefore, the interface of SiO_2_/WS_2_ is positively charged, resulting in noticeable N-type doping in the channel, which reduces the Schottky barrier for metal contact. Under negative V_gs_ irradiation, excited electrons in the dielectric layer may drift to the interface of SiO_2_/WS_2_, either neutralized with the hole or captured by the defect in the interface. In this case, electron cumulation may occur in the interface, with a large number of holes caught in the dielectric layer far from the top WS_2_. Because the doping of these holes and electrons neutralizes each other, weak P-type doping results. This work contributes to the study of the physical properties of atomically thin TMDs using electron microscopy.

Chen et al. [93] demonstrated the controllable growth of WS_2_ through CVD at different indium (In) doping concentrations. By modulating the weight ratio of the precursor (In_2_O_3_, WO_3_), the controlled doping of WS_2_ is realized. When the precursor ratio (In_2_O_3_:WO_3_) is 1:8, the PL intensity of WS_2_ is significantly enhanced, and the enhancement factor can reach ~35. With the further enlargement of doping concentration (1:5), the luminous intensity showed a downward trend, and when the precursor ratio reached 1:3, the luminescence decreased to the undoped level or below. In addition, WS_2_ FETs are regularly transformed from N-type to bipolar and finally to P-type. It shows strong N-type properties in undoped FETs. When the mass ratio of the precursor In_2_O_3_:WO_3_ is greater than 1:8, the bipolar electric transfer curve appears, and when the mass ratio of the precursor In_2_O_3_:WO_3_ is greater than 1:3, it shows a complete P-type. Since the study of optical and electrical dopants is rarely reported, this study has potential application prospects in electronics, optoelectronics, and other fields.

#### 3.3.5. Synaptic Transistors

Luo et al. [94] reported photo memristor transistors based on PbZr_0.2_Ti_0.8_O_3_ (PZT) and WS_2_ thin films. The transistor can respond to both light and electrical irritation and mimic biological light sensing and synaptic functions, providing a possible solution for neuromorphic visual synaptic devices and optoelectronic storage applications.

Chen et al. [95] reported an integrated synaptic transistor from ultra-thin ferroelectric HZO and 2D WS_2_. The transistor has an enormous storage window, showing an on/off ratio of ≈10^6^ between two current states associated with dissimilar voltage sweep orientations. At the same time, this non-volatile synaptic transistor simulates the basic functions of biological synapses with good data storage capabilities, demonstrating the potential application of neuromorphic computing applications.

#### 3.3.6. Phototransistors

Better channel control is essential to improve the performance of existing devices. Zhang et al. [96] fabricated a multi-position controllable gate WS_2_/MoS_2_ heterojunction phototransistor. Schematic and optical images of the transistor are shown in Figure 11a, b. In this transistor, the top gate can modulate the carrier transport of WS_2_ at the top of the heterojunction, and the rear gate can modulate the carrier transport of the MoS_2_ and WS_2_ regions on both sides of the heterojunction, as displayed in Figure 11c. Figure 11d shows the band diagram of the device, with internal barriers forming at the heterojunction, causing the device to exhibit rectification behavior. Figure 11e–h depicts the current-voltage characteristic curve of the transistor. In Figure 11e,f, it can be seen that only when the drain voltage (V_d_) is at the negative axis, the top-gate voltage (V_tg_) and the back-door voltage (V_bg_) can significantly modulate the drain current (I_d_). When V_tg_ = V_bg_ = 0, the transistor shows rectification behavior because of the difference of WS_2_ and MoS_2_ work function, and the rectification ratio can be modified from 1 to more than 10^4^ through V_tg_ and V_bg_. It can be observed in Figure 11g,h that both negative V_tg_ and negative V_bg_ can turn the transistor “off” with an on/off ratio of up to 1.5 × 10^7^. The device also exhibits a very low subthreshold swing due to the formation of an extra homojunction at the interface. These results will be conducive to the progress of multifunctional optoelectronic devices based on 2D materials for light detection, logic computation, and functional circuits.

#### 3.3.7. Summary

In WS_2_-based transistors, the main bottleneck is the high contact resistance between the metal and WS_2_, the Fermi-level pinning, and the Schottky barrier. Reducing the contact resistance by semi-metallic contacts is an effective method, but it comes at the cost of weakening device stability and reliability [83]. However, double-layer materials have better immunity to fabrication-induced damage or interface scattering. However, there is currently limited research on transistors based on double-layer materials, such as a lack of research on the electrical characteristics of ultra-large double-layer transistors with high saturation speeds [97]. In addition, research has shown that reducing interfacial defects, interfacial doping, and optimizing device structures, such as designing vertical field-effect tunneling transistors and edge contacts, can effectively improve Fermi level pinning and Schottky barrier [98,99,100]. Zhang et al. [101] pointed out that the thin-film resistance and contact resistance at the interface are related to the size and uniformity of the semiconductor material. Therefore, CVD is often used for transistors to produce uniform thin films with large areas. In addition, in Zhang’s study, WS_2_ transistors with a thickness of nine layers showed the best performance. Therefore, thickness and morphology have a significant impact on the application of transistors [102]. In this manuscript, we present the synthesis methods and applications of representative 2D WS_2_ with different morphologies in the form of a table for intuitive understanding. The Table 3 is as follows:

### 3.4. Photodetectors

The photodetector can change optical signals into electrical signals [103] and has great significance in the fields of national defense, environment monitoring, optical communication, and biomedicine. The lack of dangling bonds in 2D layered materials allows them to form vdW heterojunctions with other semiconductors [9]. Tungsten disulfide (WS_2_), one of the most valuable constituents of 2D TMDs, displays layer-associated band gaps ranging from 2.1 eV in monolayers to 1.4 eV in blocks, offering superb adaptability for the assembly of visible-near-infrared photodetectors [104]. Here, we mainly introduce the five categories of WS_2_-based photodetectors, including flexible photodetectors, infrared photodetectors, ultraviolet photodetectors, and wide spectrum photodetectors.

#### 3.4.1. Flexible Photodetectors

Flexible photodetectors have important research significance in medicine, wearable devices, and other fields. Under the premise of mechanical strain, maintaining high responsivity, detectivity, fast response speed, and excellent photocurrent/dark current ratio is the key to the flexible photodetector. Jang et al. [105] reported for the first time the preparation of a semi-transparent photodetector at an ultraviolet to visible wavelength by doping trifluoromethanesulfonamide graphene (TFSA-GR)/WS_2_ upright heterostructures on rigid glass and flexible plastic polyethylene terephthalate (PET) substrates, as shown in Figure 12a. In the prepared photodetectors, WS_2_ is approximately four to five layers thick, doped with graphene using TFSA to improve the graphene carrier mobility, characterized by Raman spectroscopy as p-type doping, and has a greater responsivity than undoped graphene/WS_2_ photodetectors—up to 0.14 AW^−1^. Figure 12b–d shows the band structure when the bias voltage is equal to 0, less than 0 (reverse bias), and greater than 0 (forward bias), respectively. When a reverse bias is exerted, the electrons need to surmount a higher barrier to flow to WS_2_. When forward bias is exerted, the electrons only need to surmount the lower barrier to flow toward TFAS-GR. Therefore, the current at the forward bias voltage will be larger. Figure 12e shows the repeated on/off current switching behavior at different optical powers, with a 65 μs rise time and 185 μs fall time. It can be seen from Figure 12f that the photocurrent and optical power have a good linear relationship. Figure 12g shows the relationship between transmittance and optical wavelength. In the visible range, the average transmittance is 40%, which is translucent. Figure 12h compares the repeated on/off characteristics of photocurrent density with/without mirrors. By positioning the mirror, the photocurrent density is increased from 47 μA/cm^2^ to 52 μA/cm^2^—an increase of 11%. The detector also has a self-powered function, which is generated by a built-in potential and negligible dark current at zero bias, greatly reducing losses. The detector has a maximal detectivity and EQE of 2.5 × 10^9^ cm Hz^1/2^ W^−1^ and 40%, which is better than most 2D material-based hybrids. The responsivity after 3000 bends is 88%, showing excellent transparency and flexibility. Their research demonstrates the potential of doping processes and heterostructures to improve the performance of photodetectors.

Kim et al. [106] successfully synthesized flexible WS_2_ photodetectors by RF magnetron sputtering and electron-beam irradiation. The resistance of the treated WS_2_ is reduced due to the presence of SV and atomic rearrangements. The photoresponsivity of the detector at 450, 532, and 635nm is 1.25, 1.66, and 0.53 mA W^−1^, respectively—three orders of magnitude higher than those of the deposition state. The rise time is 0.48, 0.86 and 0.82 s, and the fall time is 0.70, 0.88 and 0.79 s, respectively. The detectivity is 2.52 × 10^7^, 3.34 × 10^7^, and 1.08 × 10^7^ Jones, respectively, which is more than 10 times higher. In static and cyclic bending tests, the optical response performance remains largely unchanged (<10% loss), thus confirming the durability of the flexible device.

#### 3.4.2. Wide-Spectrum Photodetectors

Wu et al. [107] constructed a WS_2_/AlO_x_/Ge vdW heterojunction-based ultrawideband photodetector through defect engineering and interface passivation, as shown in Figure 13a. In Figure 13b, the WS_2_/AlO_x_/Ge vdW heterojunction exhibits obvious rectification behavior, which results from the interface passivation between WS_2_/Ge and AlO_x_. In addition, under the light (1550 nm, 9.0 mW/cm^2^), the device has a 0.95 mA short-circuit current and a 0.23 V open-circuit voltage. Figure 13c depicts the time–light response of the photodetector in photovoltaic mode, disclosing the stable and replicable photocurrent. The wavelength-dependent response at 42 μW/cm^2^ light intensity is shown in Figure 13d. Compared with the spectral response of the pure Ge photodetector, the WS_2_/AlO_x_/Ge photodetector has a higher optical response when the wavelength is greater than 1800 nm. The optical response characteristics of the WS_2_/AlO_x_/Ge photodetector are also affected by the light intensity. Figure 13e shows the I–V curves under 1550 nm light at different intensities. With the increase in light intensity, both short-circuit current and open-circuit voltage augment. Figure 13f shows the values of responsivity (R) and EQE at different light intensities. The values of R and EQE augment with the decrease in light intensity. In addition, the photodetector has a specific detection rate of up to 4.3 × 10^11^ Jones, which is much better than many other photodetectors, and the device has excellent device performance, such as a rise/fall time of 9.8/12.7 μs.

Jia et al. [108] demonstrated a wideband photodetector based on WS_2_/GaAs heterojunctions with Zener tunneling properties and self-actuating capabilities. The detector has a significant optical response in the range of 200~1550 nm; it also has a low noise current (59.7 pA), high responsiveness (527 mA/W), an ultra-high light/dark ratio (10^7^), a large specific detection rate (1.03 × 10^14^ Jones), utmost detection light intensity (17 nW/cm^2^), and an 80% EQE. Wu et al. [109] using the thermal decomposition method and a standard wet transfer process prepared a 2D-WS_2_/3D-pyramid Si heterojunction photodetector that was self-driven. The device exhibits favorable rectifying characteristics and larger rectifying ratios. Moreover, it is highly sensitive over an ultra-wide spectral range from 200 nm to 3043 nm, which illustrates its potential superiorities in optical communication, imaging, environment monitoring, night vision systems, and remote sensing. Kanade et al. [110] investigated a new method for the synthesis of TMDs at low temperatures, using H_2_S and Ar gas plasma to further vulcanize the precursor at relatively low temperatures (150 °C); synthesized wafer-scale MoS_2_/WS_2_ vertically stacked heterostructures; and demonstrated a photodetector based on MoS_2_/WS_2_ heterostructures. The photodetector has a maximum photoresponse of 83.75 mA/W under 660 nm illumination, an average detectivity of 2.48 × 10^7^ Jones, and an optical switching time of 60 ms, which is superior to most reported photodetectors. In addition, it also has an efficient photocurrent generation phenomenon, highlighting the advantages of an MoS_2_/WS_2_ heterostructure in advanced photodetectors. Chowdhury et al. [111] demonstrated for the first time the chemical doping and plasma-enhanced light response of WS_2_/Si heterojunctions and reported a photodetector based on the vertical WS_2_/Si heterojunction. The detector has a peak response rate of 8.0 a/W, an EQE of 200%, and a detection rate of 1.5 × 10^10^ Jones at −10 V bias, which is much higher than that of merchant silicon photodetectors. This study offers a new example of the stripping of N-type low-layer WS_2_ assisted by undoped metal nanoparticles. Nguyen et al. [112] described a fast light response-enhanced wideband photodetector composed of a WS_2_/ZnO heterostructure. The heterostructure exhibits enhanced light absorption that spans the ultraviolet light to the visible light range, which contributes to an overall improvement in light detection performance when compared to ZnO photodetectors. WS_2_/ZnO devices exhibit high performance, fast responses to ultraviolet light, a responsiveness of 2.7 AW^−1^, and a detection rate of 5.8 × 10^12^ Jones. This study offers a simple approach to improving the properties.

#### 3.4.3. Infrared Photodetectors

The lifetime and mobility of the carrier determine the performance of the infrared photodetector to some extent, but the carrier lifetime generated by infrared at room temperature is short, which reduces the efficiency of the detector. HfS_2_ has a deeper CBM and higher electron mobility, which facilitates interlayer exciton (ILE) residency when paired with WS_2_. Lukman et al. [113] designed a mid-infrared photodetector using interlayer excitons produced by WS_2_ and HfS_2_. Figure 14a is the schematic diagram of the device, and Figure 14b shows the current–voltage curve under different levels of power laser irradiation. The presence of photocurrent in the second quadrant indicates that carriers are accumulating near the heterojunction, rather than being depleted. As shown in Figure 14c, the heterojunction acts as a trap by binding ILEs around it. Trapped ILEs can borrow their oscillator strength from their surroundings, and the buildup of ILEs may have augmented the electron-hole overlap around the interface, helping to overcome the spatial indirection of the transition. Figure 14d plots the I–V curve of WS_2_/HfS_2_ excited by an infrared laser (λ_exc_ = 4.7 μm). When the drain applies a negative voltage relative to the source (V_ds_ < 0 V), the photocurrent increases—in this case, charge extraction is advantageous, as shown on the left of Figure 14e. When V_ds_ is greater than 0 V, ILEs will accumulate at the interface, resulting in more recombination and fewer extractable carriers, so the photocurrent will decrease, as shown in Figure 14e. With a modest electric field, the absorption band of the detector can be tuned to 20 µm. Owing to the distinctive band arrangement and orbital hybridization, the absorption of ILEs is enhanced, resulting in a strong light response. Figure 14f–h shows the comparison of the WS_2_/HfS_2_ heterostructure with other photodetectors. From Figure 14f, it can be seen that the ILE-enhanced absorption improves its response rate by two orders of magnitude in the nanometer low-infrared range. As illustrated in Figure 14g, the inherent dipole of an ILE, which is reactive to an externally exerted field, can be employed to dynamically adjust the detection scope and sensitivity. As illustrated in Figure 14h, the detector prepared exhibits a superior detection rate at room temperature in comparison to other infrared photodetectors. This research offers a novel perspective on the evolution of infrared detectors.

The lateral photovoltaic effect (LPE) plays an integral role in the field of photodetectors because of its strong light sensing and positioning ability. WS_2_ is a promising candidate for improving the LPE. Nevertheless, its inherent band gap restricts the application. To solve this problem, Zheng et al. [114] proposed a photodetector based on a polymeric ferroelectric material P(VDF-CTFE) and WS_2_/Si heterostructure, which can change the potential barrier and band gap of the WS_2_/Si structure through the polarization of ferroelectric, thereby improving the lateral photovoltaic effect and the performance of the detector. The authors confirmed through experiments and calculations a ferroelectrically tuned near-infrared LPE with a high sensitivity, where the polarizing electric field enhances the LPE intensity and response speed and extends the response wavelength of the LPE to the near-infrared region of 850–1550 nm. Özdemir et al. [115] reported an infrared photodetector based on WS_2_ and PbS colloidal quantum dots. The response rate is 1400 A/W and the detection rate is 10^12^ Jones, highlighting the importance of selecting the right TMDs channel.

#### 3.4.4. Ultraviolet Photodetectors

Ultraviolet photodetectors can transform ultraviolet signals into electrical signals and play an important part in security communication, chemical analysis, and other fields. Two-dimensional/three-dimensional hybrid heterostructures can provide greater light absorption and light-generated carrier separation efficiency, so constructing mixed-dimensional heterojunctions is an efficient strategy to improve the properties of two-dimensional photodetectors. Zhao et al. [116] successfully synthesized a large area of WS_2_ film using the vulcanization method and prepared a WS_2_/GaN self-powered ultraviolet photodetector by wet transfer, as shown in Figure 15a. The detector has a rectification ratio greater than 10^5^ at 5 V, as indicated in Figure 15b. Figure 15c indicates the I–V characteristics of the photodetector under 215, 245, 300, and 375 nm illumination, and it can be found that the WS_2_/GaN heterojunction has a significant photovoltaic effect. At 375 nm light, the short-circuit current and open-circuit voltage are 3.18 mA and 1.73 V, respectively, as indicated in Figure 15d. Figure 15e indicates the time-resolved optical response characteristics of the device with zero bias. When the incidence ultraviolet light is repeatedly switched on and off, the current switches between high-current and low-current states, which has strong repeatability. Figure 15f displays the response spectrum of a heterojunction photodetector sensitive to ultraviolet light. In this detector, the photocurrent increases as the light intensity increases, and the I_SC_ can be regulated by the light intensity. In addition, the photodetector has a specific detection rate (4 × 10^14^ Jones) and a fast response rate (7.3 μs/0.42 ms).

Shelke et al. [117] successfully prepared an WS_2_/RGO ultraviolet photodetector through a hydrothermal reaction. In the test of ultraviolet light sources with different power densities, the photocurrent of WS_2_ will decrease with the increase in light power and thus show negative photoconductivity characteristics, while the photocurrent of WS_2_/RGO will increase with the increase in light power and show positive photoconductivity characteristics. The detector has an optical responsiveness of 80 mAW^−1^ and can maintain high sensitivity for a long time under different light intensities of ultraviolet light and response times, and recovery times of 48 and 85 s, respectively. Their findings provide a reference for the study of room-temperature ultraviolet photodetectors.

#### 3.4.5. Summary

To date, various two-dimensional van der Waals heterostructures have been constructed for photodetectors. These devices typically operate in photovoltaic mode with integrated electric fields, resulting in the effective suppression of dark current, separation of photocarriers, and promotion of detection. Nevertheless, the occurrence of interface recombination under the Shockley–Read–Hall or Langevin models results in considerable photocarrier losses and a deficiency in collection efficiency [118]. The study by Luo et al. [119] demonstrates that heterojunctions with I-type band arrangements can suppress interface recombination and transport capture effects, introduce effective photoconductivity gains, and enhance the collection efficiency of photo carriers. However, due to the limited cross-section of photon absorption in the thickness of atomic-thin layers, the generation of photo carriers is constrained. The restricted absorbance of WS_2_ results in low responsiveness, which may impede their performance due to weak light–matter interactions [120,121]. Fang et al. enhanced the photoresponse by combining WS_2_ with perovskite to form a heterojunction, while Zhang et al. [122] effectively adjusted the charge carrier properties of WS_2_ by applying pressure, greatly enhancing its light absorption ability. In general, single-layer or few-layer WS_2_ nanosheets are optimal candidates for high-performance visible-light-range photodetectors due to their robust electron-hole confinement, exceptional light absorption, and rapid response time [123]. The mobility of double-layer WS_2_ is twice that of single-layer WS_2_, and due to its higher thickness, it exhibits enhanced light absorption, emphasizing the importance of using double-layer WS_2_ [15]. The synthesis methods and applications of representative 2D WS_2_ with different morphologies are presented in this manuscript in the form of a table for intuitive understanding. The Table 4 is as follows:

### 3.5. Field Emission

Field emission refers to the process of electrons leaving metals or semiconductors under the action of electric fields [124]. It is an essential occurrence in materials science and has a wide range of applications in electron microscopy, microwave power amplifiers, vacuum electronics, and solar farms [125].

#### 3.5.1. Field Emission Devices

Wang et al. [14] prepared WS_2_/diamond heterojunctions by electron-beam vapor deposition (EBVD), as shown in Figure 16b,d. A WS_2_ thin film with a thickness of 10 nm is uniformly deposited on diamond particles. In the diamond/WS_2_ heterojunction, electrons can be accelerated along the graphite channels in the polycrystalline diamond grain boundaries, ultimately reaching the interface between diamond and WS_2_. The rough surface of the diamond results in a notable field enhancement effect at the interface between WS_2_ and the diamond. Furthermore, the barrier height and width at the interface between diamond and WS_2_ are relatively modest, facilitating electron penetration into the WS_2_ nano-layer through the internal field emission effect. Upon entering the WS_2_ layer, electrons can continue to maintain a high energy state until they reach the interface between WS_2_ and the vacuum, thereby increasing the field emission current density. Figure 16a,c show the electron transfer diagram of the diamond/WS_2_ composite film FE device and the corresponding band structure diagram of the diamond/WS_2_ composite film FE device.

Loh et al. [126] synthesized WS_2_ with vertically growing nano-petals and nano-hairs on CNFs by RF magnetron sputtering. As illustrated in Figure 17a–h, with the increase in deposition time, the roughness of CNFs increases significantly, and WS_2_ gradually transitions from a nano-flake to a triangular flake forest with a more pronounced, narrower, and denser appearance. The triangular flake forest provides additional sites for electron emission, resulting in enhanced field emission enhancement factors for all samples compared to the original CNFs and WS_2_. The highest values are observed in the samples with deposition times of 40, 50, and 60 min. Notwithstanding the aforementioned observations, it is important to note that the samples with the highest values are not without their limitations. The poor electrical contact and high contact resistance between the nano-petals and nano-bristles result in a significant voltage drop between these components under large emission currents, which ultimately leads to current saturation. Furthermore, a thicker coating may also result in greater electron scattering, thereby reducing the efficiency of electron transmission in the nano-sheet. Consequently, these samples exhibit the lowest conduction field, as illustrated in Figure 17i,j. In all the samples, the on-voltage of the sample at 30 min is the lowest. Furthermore, the linear relationship depicted in Figure 17j indicates that the field emission of the sample is contingent upon the quantum tunneling effect.

Grillo et al. [124] reported the preparation of WS_2_ nanotubes and experimental studies of their electrical and electromechanical properties. Before electron-beam irradiation, the field enhancement factor β is about 65, and the field emission current density is ≈6 kA cm^−2^. After the irradiation time is 30min, the field emission current density grows to a maximal value of ≈600 kA cm^−2^. However, the field enhancement factor β was reduced by about 50. This is due to Joule heating removing the surface adsorbent at the tip of the nanotubes during the first scan. In general, these adsorbs act as convergence points for electric field lines. This further increases the local electric field. The primary effect of electron irradiation is the physical modification of the contact area due to the presence of some residual hydrocarbon molecules. These molecules break down when irradiated by an electron beam to produce a carbonaceous material that acts as an adhesive, strengthening nanotube-to-tip contact and reducing contact resistance. In addition, the resistivity of the nanotubes increases exponentially with the applied axial tensile stress, which proves the applicability of a single multi-wall WS_2_ nanotube as a piezoresistive strain sensor.

#### 3.5.2. Summary

Although TMDCs have excellent electronic properties and environmental stability, there are relatively few field emission studies based on WS_2_ [126]. The application of WS_2_ in field launch is often limited by inherent defects. One potential solution to this issue is to combine different materials into composite materials or heterostructures. This approach can enhance existing properties or induce other properties that may be challenging to achieve in homogeneous systems. For instance, heterojunctions with diamond or CNFs can be formed. Field emission devices based on TMDCs are typically composed of micro peak arrays, nanotubes, or 1D nanowires. We presented the synthesis methods and applications of 2D WS_2_ with different morphologies in this manuscript in the form of a table for intuitive understanding. The Table 5 is as follows:

### 3.6. Memory Devices

As a vital part of the processing unit, memory has become the most widely used semiconductor product. The atomic thickness and favorable electrical properties of 2D nonvolatile memory (NVM) devices make them ideal for use in memory devices, and they have attracted considerable research attention for high data storage capacity and fast read and write speeds [127]. Van der Waals heterostructures formed from 2D materials can offer unlimited possibilities in the field of storage.

#### 3.6.1. Non-Volatile Memory Devices

Zhou et al. [128] implemented a quasi-nonvolatile memory with ultra-fast read and write speeds. The rectifier ratio of the device does not disappear immediately after removing the gate voltage. The rectifier ratio is still as high as about 10^5^ after 1 h, and the device still shows rectification characteristics even after 2 days. The device writes “1” very quickly because the electrons in the graphene layer can directly tunnel to the WS_2_ layer. The direction of the V_ds_ applied when reading the data is consistent with the direction of the electric field established in the junction region, and the data can be read quickly. The positive pulse added to V_g_ makes the holes in the graphene layer directly tunnel into the WS_2_ layer to recombine with the electrons, which can complete rapid erasure. In addition, the device can complete detection and stored procedures in less than 1 s. These results provide a possible direction for the next generation of low-power high-speed semi-floating gate FET applications.

Das et al. [129] prepared a WS_2_-based nonvolatile memory by CVD. The device transitions to a low-resistance state (LRS) at +1.62 V and to a high-resistance state (HRS) at −1.45 V. The SET and RESET voltages indicate that digital information can be written and deleted from the storage device, respectively. The maximum current ratio between the OFF state and the ON state is approximately 10^3^. Furthermore, following the bistable transition from the OFF state to the ON state, the device remains in the ON state even after the removal of any bias voltage, thereby demonstrating the non-volatility of the memory cell. In light of the aforementioned findings, Figure 18 illustrates the underlying mechanism behind the resistance switch behavior observed in the device. In the case of WS_2_ thin films, the contact between adjacent crystallites results in the formation of a large number of grain boundaries, as illustrated in Figure 18a. This process results in the formation of a high junction barrier. Nevertheless, the application of an electric field results in the injection and accumulation of charges. As illustrated in Figure 18b, the elevated concentration of charge carriers within the nano-WS_2_ network gives rise to a diminution of the junction barrier height. This, in turn, facilitates the electron tunneling and transmission processes. This results in the formation of a conductive path along the grain boundary of the nanoparticles, which sets the device to the LRS, as shown in Figure 18c. As shown in Figure 18d, by applying a certain reverse electric field, the conductive channel can be broken, and the device can enter the HRS. At low voltage, the conduction mechanism is ohmic, while at high voltage, the trap-controlled space-charge-limited current mechanism is dominant, thus exhibiting excellent retention and durability.

Ye et al. [130] systematically investigated the long-lived interlayer exciton/valley polarization hysteresis in WS_2_/WSe_2_ heterostructures and successfully designed a non-volatile valley addressable memory. Due to the type II band alignment of the heterostructure and the synergistic blocking effect of the chemical barrier between WSe_2_ and O_2_/H_2_O, the data of this memory can be retained for at least 60 min, which holds great promise for non-volatile valley addressable memory. Compared with similar memory devices, the memory also has the advantages of low power consumption and short switching time. Siao et al. [131] report a diffusion-controlled multielement doping (MED) CVD route with a large memory window in the doped WS_2_ FET transfer curve. The memory window is attributed to the presence of impurity states, polar molecules, and sulfur vacancies in the vicinity of the conduction band minimum of metal-doped WS_2_. The wide range of current platforms in the I_DS_-V_GS_ curve has information storage capabilities that can be used to distinguish drain currents in the “ON” and “OFF” logic switching states in response to the gate voltage, and it is worth noting that the memory effect here is permanent, unlike the hysteresis loops caused by surface adsorption or semiconductor/dielectric interface capture states. The transition between the program state and the erase state is achieved by applying a voltage pulse to the backdoor electrode to charge or discharge the impurity state below the conduction band of the MED WS_2_. After the erase state lasts for 104 s, the threshold voltage varies between 12.7 and 10.2 V. If the reference erase state is characterized as Vth ≈ 0 V, the charge loss after 10 years can be estimated to be about 20%. This technique is suitable for the application of non-volatile storage technology. Previous studies have shown that shallow-trap energy levels can easily lead to storage state ambiguity, and controlling the distribution density and size of nanocrystals is also an important issue. Zhu et al. [132] designed and fabricated a WS_2_-based nonvolatile memory using Ta_2_O_5_ as the charge-trapping medium. Thanks to the better thermal stability, higher dielectric constant, deeper charge levels, and relatively smaller band gap of Ta_2_O_5_ dielectric, they achieved low operating voltage, fast P/E speed, and good reliability.

An NVM device architecture that improves long-term charge retention with minimal charge loss without affecting the memory window and extinction ratio to fulfill future standards for 2D-NVM devices in applications has become an urgent need. Siao et al. [127] first reported the application of WSe_2_/WS_2_ vdW heterogeneous structure band offset engineering to NVM. The proposed NVM architecture resembles a three-terminal FET. The memory characteristics are controlled only by the reverse voltage (V_bg_) without any additional control grid. The device has the smallest charge loss (7%) after 10 years of retention and opens a 34 V memory window while maintaining a 10^6^ extinction ratio.

#### 3.6.2. Summary

The switching between high-resistance and low-resistance states of devices can result in high power consumption and hinder the further application of non-volatile memory. Complementary inverters are essential components in the construction of TMD-based logic circuits, and they can also reduce power consumption. Shen et al. proposed the development of complementary inverters with integrated logic and storage functions, which is of great significance [133]. When WS_2_ forms non-volatile memory with different materials, they exhibit significant performance differences. Furthermore, it is essential to optimize the layer morphology and thickness, including the working voltage, durability, retention force, on/off ratio, and other parameters, to obtain the optimal parameters [134]. The manuscript presents the application of 2D WS_2_ with representative different morphologies in the form of a table, which provides an intuitive understanding of the relevant parameters. The Table 6 is as follows:

## 4. Optoelectronic Applications of Other TMD Materials

Transition-metal disulfide compounds have similar properties. The optoelectronic applications based on MoS_2_, WSe_2_, ReS_2_, and PtS_2_ have also received considerable attention. Single-layer MoS_2_ has high absorption and strong exciton generation ability, but unfortunately, it cannot be detected by polarized light. Lei et al. [135] successfully extended the photosensitivity of MoS_2_ in the visible to near-infrared region under uniaxial stress. MoS_2_ nanoflowers have a large specific surface area and are excellent sensing materials. Bharathi et al. [136] used Ni-doped MoS_2_ nanoflowers to increase the edge active sites and improve the sensing capability for NO_2_. The band gap of MoS_2_ can be adjusted from 1.2 eV to 1.9 eV. As the size continues to shrink, it also exhibits size-dependent optical properties, providing greater flexibility for LED design [137]. Pan et al. [138] achieved a self- powered photodetector by combining MoS_2_ and InSe_2_ into a type II heterojunction, which can display light response even under zero external bias. Single-layer WSe_2_ crystals exhibit excellent surface properties, such as minimal surface roughness, defect states, natural oxides, and dangling bonds, resulting in excellent carrier mobility and improved channel charge inflection points in FETs [139]. Islam et al. [140] doubled the drain current of WSe_2_-based FETs by low-temperature annealing. In addition, the lower bandgap, effective electron mass, and electron affinity of WSe_2_ indicate that WSe_2_ will be a much better field emitter than MoS_2_. Bartolomeo et al. [141] observed the first gate-controlled field emission current from WSe_2_. ReS_2_ is a novel member of the TMDC family of materials. Due to its weak interlayer coupling and each layer being a decoupled monolayer, ReS_2_ has a direct bandgap independent of the number of layers and strong anisotropy, making it suitable for designing efficient electronic and optoelectronic flexible broadband detection devices with novel concepts. Lin et al. [142] enhanced the light scattering effect by combining ReS_2_ and GaN. Du et al. [143] relied on photoheating to induce changes in the conductivity of ReS_2_ instead of interband light excitation, making it possible to detect sub-bandgap photons. Unlike other TMD materials, PtS_2_ has a stable 1T phase (octahedral structure) and low-temperature dependence. Therefore, PtS_2_ is expected to be used in high-performance optoelectronic devices with high operating temperatures. Zhang et al. [144] developed a large-area uniform PtS_2_ photodetector, which has the potential to achieve infrared imaging and infrared sensing under extreme conditions. Liu et al. [145] constructed 1T-PtS_2_/MCN composites and improved their photocatalytic performance through defect engineering modification.

## 5. Conclusions and Outlook

Recent research has significantly advanced our understanding of the potential applications of WS_2_-related nanostructures in a range of optical and electrical components, including LEDs, sensors, FETs, photodetectors, field emission devices, and memory devices. Although WS_2_-based devices have demonstrated favorable characteristics in a variety of optoelectronic applications, it is also the objective to identify novel fabrication pathways and electrode types for these devices, to enhance future optical and electrical properties. Researchers are still attempting to develop a novel variety of morphologies for WS_2_ nanostructures, optimize preparation schemes, form heterojunctions with distinct substrates, and modify WS_2_ through physical methods to achieve more stable performance, which could be a fruitful strategy for various photoelectrical applications. For instance, the WS_2_/Ge heterojunction treated by defect engineering and interface passivation can enhance the response rate and expand the response spectrum of the device. The air stability of the device can be enhanced by treating WS_2_ nanosheets with UVO. Furthermore, different doping processes can have a significant impact on the photoelectrical performance of the material. To gain a deeper understanding of the physical transport mechanisms of the device, it is crucial to apply more correlative semiconductor theories and computational models for carrier transport studies, with a particular focus on the evolution of WS_2_ devices. The optimization of heterojunctions through the use of van der Waals forces necessitates the implementation of systematic theoretical and experimental studies.

There are few relevant studies on WS_2_ related to nanomaterials and 2D materials under extreme conditions such as high temperature and high pressure. The stability and lifetime of optoelectronic devices under extreme conditions have been critical issues. Under high-temperature conditions, the number of intrinsic carriers generated by thermal excitation will increase, which will have a certain impact on device accuracy. Moreover, high temperature may also lead to changes in the carrier transport mechanism, which probably enhances the photoelectric performance of WS_2_ heterojunction-related optoelectronic devices. Therefore, further research on behavior, device design, and optimization are required to explore WS_2_ nanostructure-based optoelectronic devices under extreme conditions.

Diamonds exhibit a wide forbidden band, high thermal conductivity, a low background current, physical and chemical stability at room temperature, and other desirable properties that make them an excellent p-type conductive material in the B-doped condition for high-voltage and high-power optoelectronic devices. In comparison to conventional heterojunction structures, the van der Waals heterojunction composed of WS_2_ and diamond may offer the potential to overcome the limitations of lattice matching and eliminate the effect of high temperature on the lattice matching. The incorporation of a WS_2_ thin-film layer into a diamond thin-film electroluminescent device allows the WS_2_ thin-film layer to function as an electron and hole transport layer, thereby enhancing the carrier transport efficiency of the device. This, in turn, greatly boosts the optoelectrical properties of the device. In essence, the fabrication of WS_2_/diamond heterojunctions offers the potential to enhance the performance of LEDs, sensors, FETs, photodetectors, field emission devices, and memory devices in extreme environments.

## Figures and Tables

**Figure 1 molecules-29-03341-f001:**
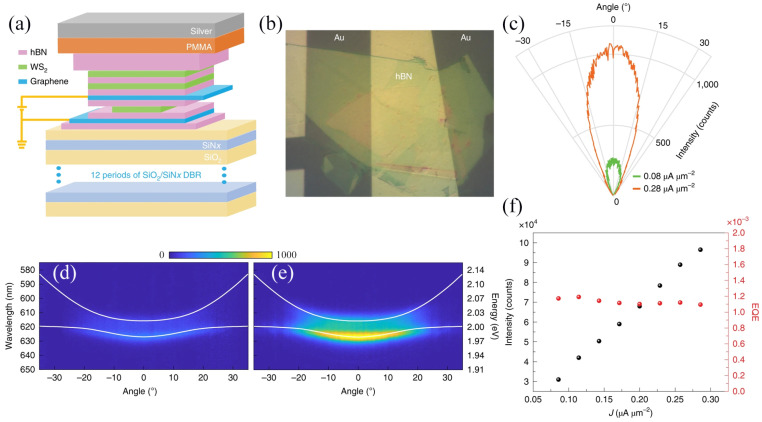
(**a**) Schematic of the device. (**b**) Optical image of the vdW heterojunction taken before top reflector growth. (**c**) Polar plots at various current densities. (**d**) Polariton dispersion from EL at an injection current of 0.08 µA·µm^−2^. (**e**) Polariton dispersion from EL at a current injection of 0.28 µA·µm^−2^. (**f**) The black curve is a function of the integrated EL intensity and current density. The red curve is the curve between EQE and current density as a function of current density. Reproduced with permission from Ref. [36]. Copyright 2019 Springer Nature.

**Figure 2 molecules-29-03341-f002:**
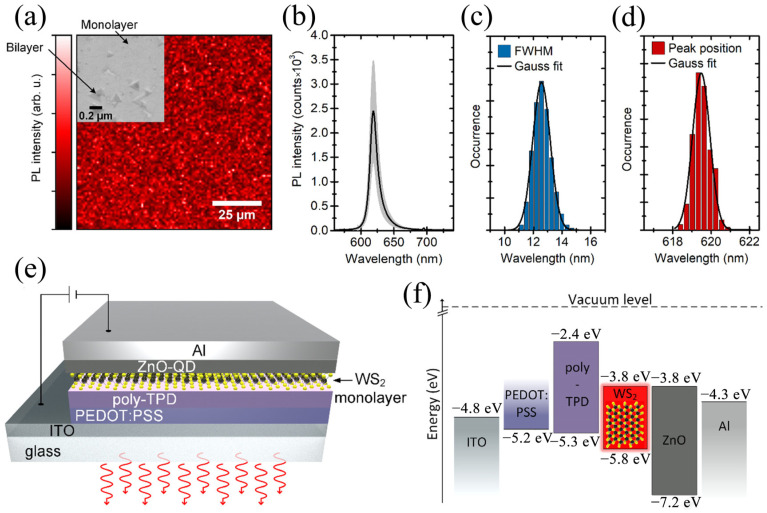
(**a**) PL peak intensity map of a monolayer of WS_2_ in which WS_2_ was grown on sapphire. A 532 nm laser was used during testing and the power density was 60 kW/cm^2^. The inset presents an SEM image of the WS_2_. (**b**) Average PL spectra, 10,000-point sites measured, gray represents standard deviation. (**c**) Histogram of FWHM distribution with an average width of 12.6 ± 1.2 nm. (**d**) T Histogram of peak positions with an average position of 619.5 ± 0.9 nm. (**e**) The device structure. (**f**) Energy level diagram. Reproduced with permission from Ref. [38]. Copyright 2019 American Chemical Society.

**Figure 3 molecules-29-03341-f003:**
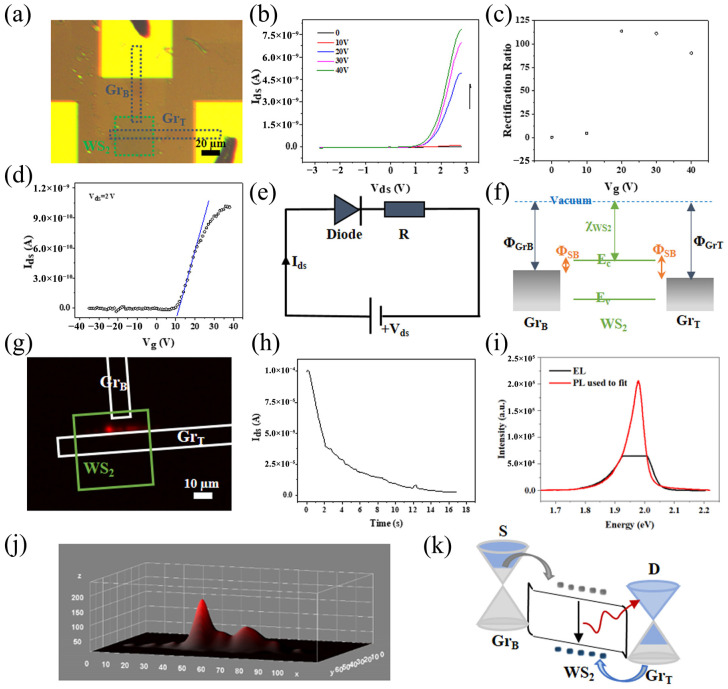
(**a**) Optical image. (**b**) Output *I*_ds_–*V*_ds_ curves under different back gates. (**c**) Rectification ratio at *V*_ds_ = ±3 V under different back gates in (**b**). (**d**) Transfer curve of the device. (**e**) Equivalent circuit model of the device. (**f**) Band diagram of the device. Φ_SB_ represents the SBH. Φ_GrB_ and Φ_GrT_ denote the work function of GrB and GrT, respectively. χ_WS2_ denotes the electron affinity of WS_2_. (**g**) Optical image characterizing red emission. (**h**) Output *I*–*T* curve of the device. (**i**) EL spectra of another device tested under the same conditions. Since the signal intensity was too high, it was fitted by PL spectroscopy. (**j**) Intensity profile of the red emission in (**g**). (**k**) Energy band diagram of the device in the EL state. The gray spheres represent electrons, and the blue represents holes. Reproduced with permission from Ref. [39]. Copyright 2023 American Chemical Society.

**Figure 4 molecules-29-03341-f004:**
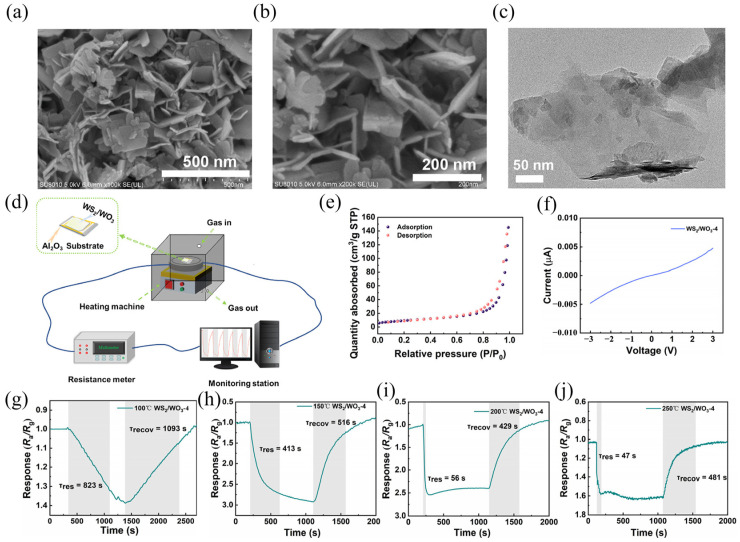
(**a**,**b**) SEM. (**c**) TEM. (**d**) Schematic diagram of the acetone induction measurement system. (**e**) Nitrogen sorption isotherms of WS_2_/WO_3_-4. (**f**) Current–voltage curve of the WS_2_/WO_3_-4 sensor. (**g**–**j**) Response and recovery plots of WS_2_/WO_3_-4 to 20 ppm acetone at different operating temperatures. Reproduced with permission from Ref. [49]. Copyright 2022 American Chemical Society.

**Figure 5 molecules-29-03341-f005:**
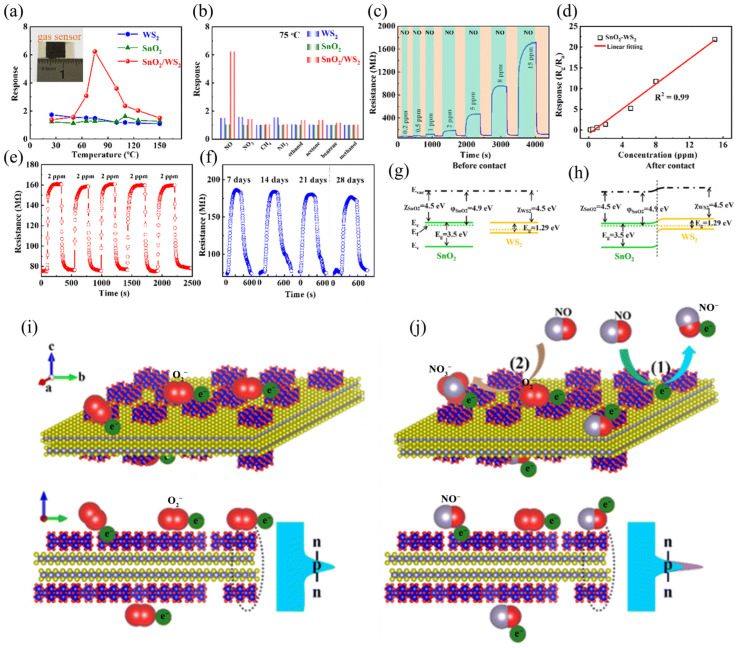
(**a**) S_r_ values of WS_2_, SnO_2_ and SnO_2_/WS_2_ sensors at various temperatures, with the concentration of NO equal to 5 ppm during testing. The inset is a photo of the gas sensor. (**b**) Selectivity test of WS_2_, SnO_2_ and SnO_2_/WS_2_ sensors when the temperature is equal to 75 °C, where the gas concentration is equal to 5 ppm. (**c**) Dynamic resistance variation curve of SnO_2_/WS_2_ sensor with NO concentration where temperature is equal to 75 °C. (**d**) S_r_ value of the sensor versus NO concentration curve. (**e**) Repeatability test of the gas sensor response to NO where the temperature is equal to 75 °C and the gas concentration is always 2 ppm. (**f**) Long-term stability testing of gas sensors in NO environments where the temperature is equal to 75 °C and the gas concentration is consistently maintained at 2 ppm. (**g**) Energy band structures before SnO_2_ and WS_2_ form a heterojunction. (**h**) Energy band structure after heterojunction formation of SnO_2_ and WS_2_. (**i**) Schematic representation of SnO_2_/WS_2_ interface and energy barriers before NO adsorption. (**j**) Schematic representation of SnO_2_/WS_2_ interface and energy barriers after NO adsorption. Reproduced with permission from Ref. [51]. Copyright 2023, Royal Society of Chemistry.

**Figure 6 molecules-29-03341-f006:**
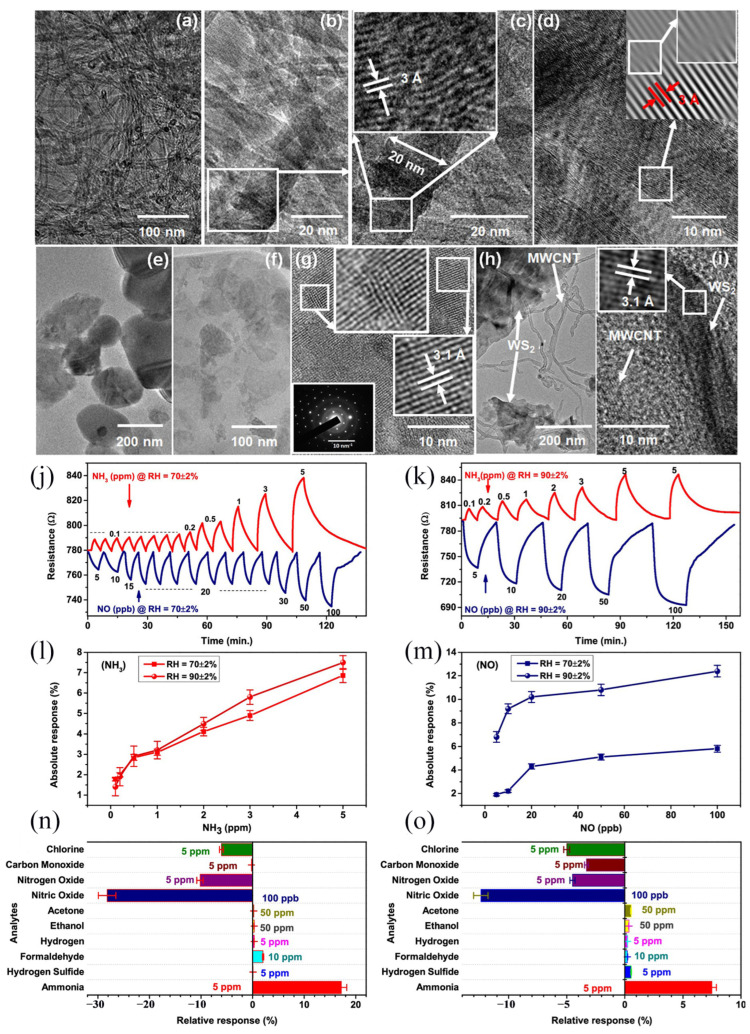
(**a**) TEM image of MWCNTs. (**b**,**c**) HRTEM images of the same sample showing MWCNTs, and (**d**) HRTEM image displaying interplanar spacing from MWCNTs. (**e**,**f**) TEM images of WS_2_ nanosheets. (**g**) Filtered images of nanosheets from panel (**f**). Inset includes SAED pattern from WS_2_ nanosheets. (**h**) TEM images of WS_2_/MWCNT composites. (**i**) HRTEM images of WS_2_/MWCNT composites. (**j**) Response of device D2 to NH_3_ and NO at 70 ± 2% relative humidity. (**k**) Response of device D2 to NH_3_ and NO at 90 ± 2% relative humidity. (**l**,**m**) Comparison of absolute relative responses corresponding to various concentrations. (**n**,**o**) Selectivity testing of D1 and D2 devices at 70 ± 2% relative humidity at 16 °C. Reproduced with permission from Ref. [58]. Copyright 2022 American Chemical Society.

**Figure 7 molecules-29-03341-f007:**
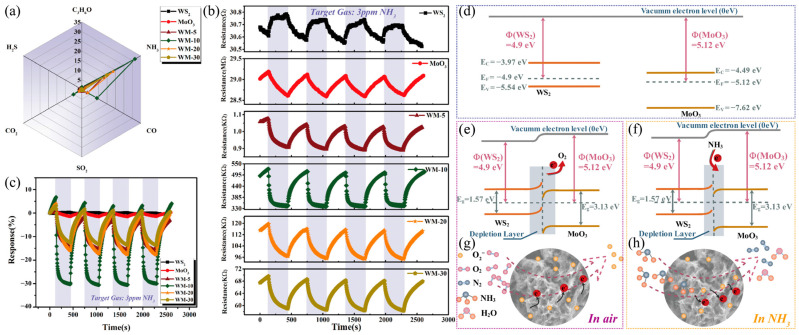
(**a**) Gas selectivity test at room temperature where the gas concentration is maintained at 3 ppm. (**b**) Reproducibility of each sensor at 3 ppm NH_3_ exposure. (**c**) Dynamic response of each sensor at exposure to 3 ppm NH_3_. Energy band diagram of WS_2_/MoO_3_ hybrids (**d**) before contact and after contact (**e**) in air and (**f**) in NH_3_. Diagram of gas sorption of WS_2_/MoO_3_ hybrids (**g**) in air and (**h**) in NH_3_. Reproduced with permission from Ref. [59]. Copyright 2022 American Chemical Society.

**Figure 8 molecules-29-03341-f008:**
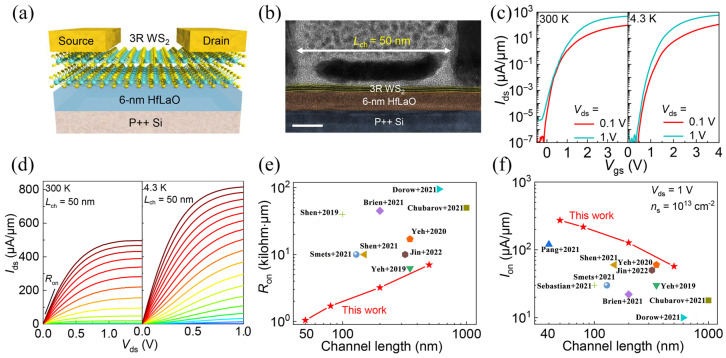
(**a**) Schematic diagram of the device, wherein the insulator HfLaO has a thickness of 6 nm. (**b**) TEM image of the device. (**c**) Transmission characteristics of the device at temperatures equal to 300 or 4.3 K. (**d**) The output characteristics of the device at temperatures of 300 or 4.3 K. Benchmark of R_on_ (**e**) and I_on_ (**f**) as a function of channel length for CVD WS_2_ transistors at room temperature. The carrier density is fixed at n_s_ = 10^13^ cm^−2^ [73,74,75,76,77,78,79,80,81,82]. Reproduced with permission from Ref. [72]. Copyright 2023 American Association for the Advancement of Science.

**Figure 9 molecules-29-03341-f009:**
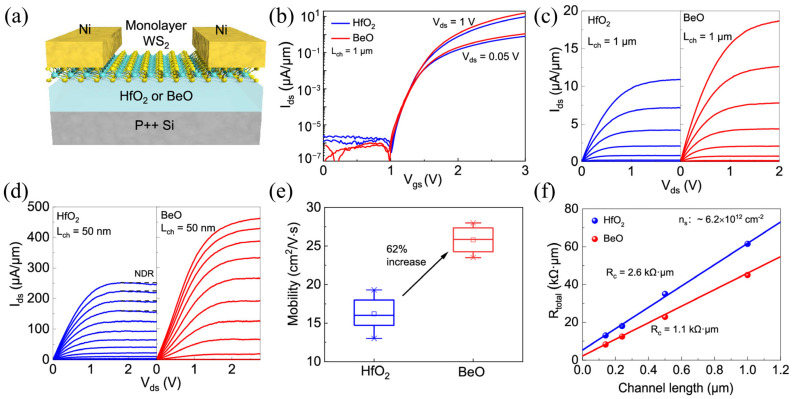
(**a**) Schematic of the single-layer WS_2_ back-gate FET. (**b**) I_ds_–V_gs_ transfer curves for transistors employing HfO_2_ or BeO dielectrics. (**c**) The corresponding I_ds_–V_ds_ output characteristics of the transistors. where L_ch_ is equal to 10 μm. (**d**) The corresponding I_ds_–V_ds_ output characteristics of the transistors. Where L_ch_ is equal to 50 nm. (**e**) Statistical distribution of mobility of FETs based on HfO_2_ or BeO dielectrics. (**f**) TLM resistors for FETs based on HfO_2_ and BeO dielectrics. Reproduced with permission from Ref. [84]. Copyright 2022 American Chemical Society.

**Figure 10 molecules-29-03341-f010:**
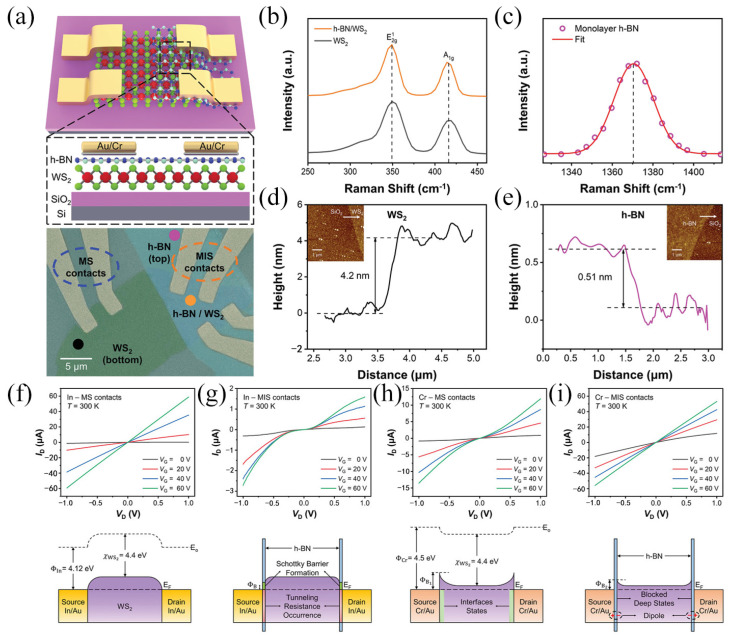
(**a**) Schematic and false-color SEM of the MIS structure. (**b**) Raman spectra of h-BN/WS_2_ heterostructure and few-layer WS_2_. (**c**) Raman spectrum of h-BN. (**d**) AFM of WS_2_. (**e**) AFM of hBN. (**f**–**i**) The above figure shows the I_D_-V_D_ curves of MS and MIS structures and In and Cr electrodes at various gate voltages and temperatures of 300 K. The following figure is the corresponding energy band diagram. Reproduced with permission from Ref. [89]. Copyright 2022 John Wiley and Sons.

**Figure 11 molecules-29-03341-f011:**
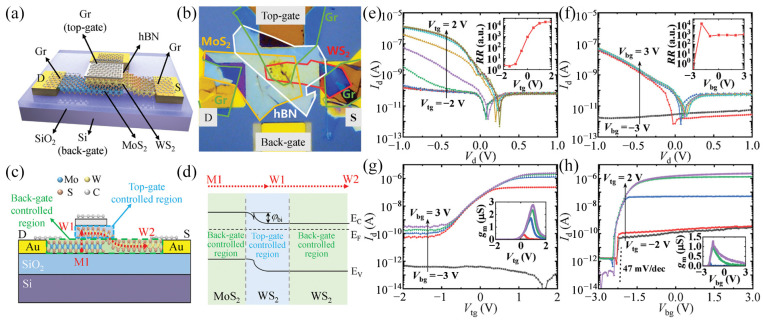
(**a**) Schematic diagram of the phototransistor. (**b**) Optical image of the phototransistor. (**c**) Schematic cross-section of the phototransistor. (**d**) At 0 V for both V_tg_ and V_bg_, the equilibrium energy band diagram of the device. (**e**) Output characteristics at various V_tg_ when V_bg_ is 0 V. The inset shows the curve of RR as a function of V_tg_. (**f**) Output characteristics at various V_bg_ when V_tg_ is 0 V. The inset shows the curve of RR as a function of V_bg_. (**g**) Transfer curves of V_tg_ at different V_bg_. The inset shows the relationship between span-to and V_tg_. (**h**) Transfer curves of V_bg_ at different V_tg_. The inset shows the relationship between span-to and V_bg_. Reproduced with permission from Ref. [96]. Copyright 2022, John Wiley and Sons.

**Figure 12 molecules-29-03341-f012:**
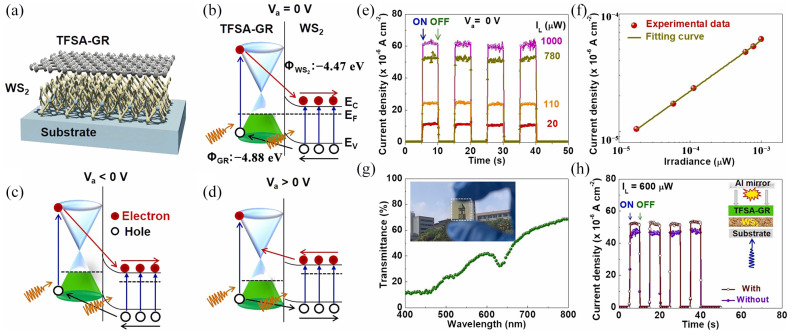
(**a**) Structural diagram of TFSA-GR/WS_2_. (**b**) Energy band diagram at bias equal to 0V. (**c**) Energy band diagram at bias less than 0 V. (**d**) Energy band diagram at bias greater than 0V. In the figure, ϕ_WS2_ and ϕ_GR_ are the work functions of TFSA-GR and WS_2_. (**e**) Repeated switching behavior of current density at various optical powers with bias voltage equal to 0 V. (**f**) Variation curve of current density with light intensity. (**g**) Spectral transmittance of PD on glass. The inset shows a photo. (**h**) Repeated switching behavior when the bias voltage is equal to 0 V. Brown represents having an aluminum mirror and purple represents no aluminum mirror, where the illumination is 600 μW. The illustration in the upper right corner shows the testing principle. The illustration in the upper right corner shows the testing principle. Reproduced with permission from Ref. [105]. Copyright 2022, Elsevier.

**Figure 13 molecules-29-03341-f013:**
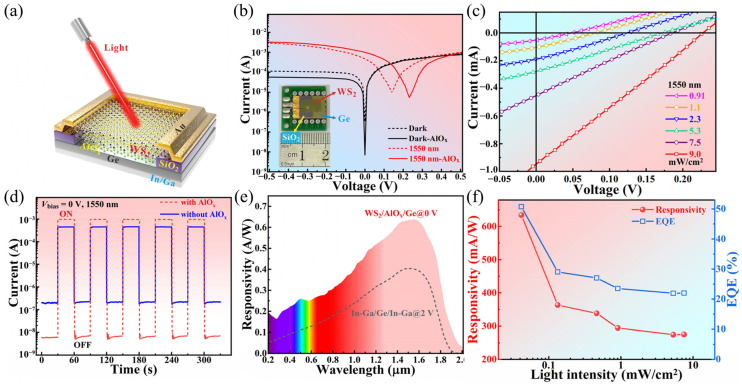
(**a**) Schematic diagram of the WS_2_/AlO_x_/Ge device. (**b**) I–V characteristics of WS_2_/Ge devices and WS_2_/AlO_x_/Ge devices in the dark and under 1550 nm illumination. The inset presents photos of the devices. (**c**) Characterization of the optical response of the device at a bias voltage equal to 0V, where the illumination is 1550 nm. (**d**) Spectral photoresponse of devices composed of WS_2_/AlO_x_/Ge and devices composed of pure Ge. (**e**) I–V characteristics of devices composed of WS_2_/AlO_x_/Ge at different light intensities at the same wavelength (1550 nm). (**f**) The red curve is the variation of R with light intensity and the blue curve is the variation of EQE with light intensity. Reproduced with permission from Ref. [107]. Copyright 2021, American Chemical Society.

**Figure 14 molecules-29-03341-f014:**
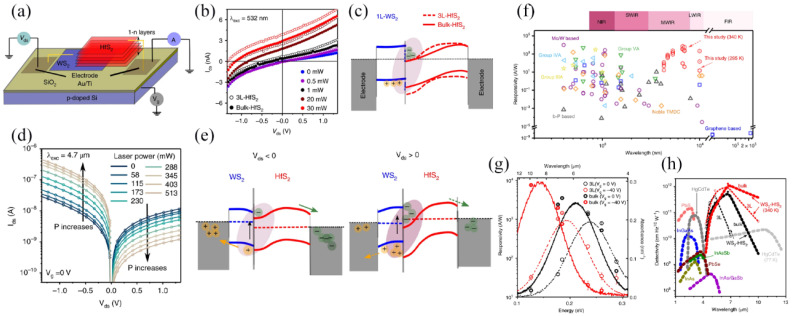
(**a**) Schematic diagram of the device. (**b**) I–V curves of WS_2_/3L HfS_2_ and WS_2_/bulk HfS_2_ at multiple excitation powers. (**c**) Energy band bending diagram. (**d**) I–V characteristics of WS_2_ devices and 3L HfS_2_ devices at different excitation powers, where λ is 4.7 μm and V_g_ is 0 V. (**e**) Schematic diagram of energy band and charge extraction of WS_2_ and HfS_2_ when Vds is greater than or less than 0V, with arrows representing the direction of carrier movement. (**f**) Comparative analysis of the responsivity of the device consisting of WS_2_/HfS_2_ with other photodetectors. (**g**) Responsiveness of the WS_2/_HfS_2_ device when V_g_ is 0 V or 40 V where V_ds_ is 1.5 V and P_device_ is 0.5 nW. (**h**) Specific detectivity as a function of wavelength for WS_2_/HfS_2_ devices and the commercially available photodetectors at room temperature. Reproduced with permission from Ref. [113]. Copyright 2020 Springer Nature.

**Figure 15 molecules-29-03341-f015:**
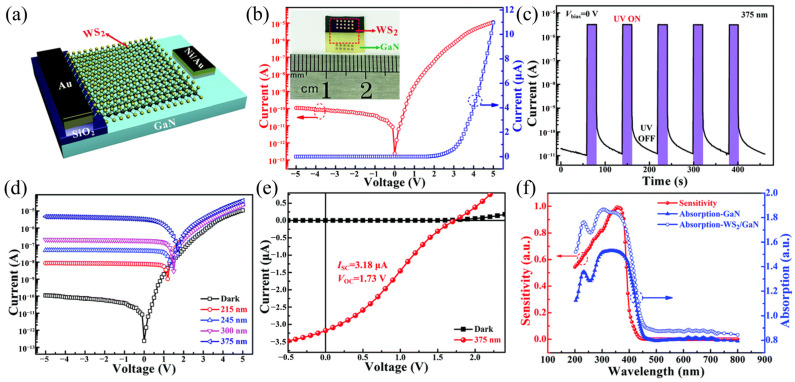
(**a**) Schematic diagram of the device. (**b**) The I–V characteristics of the device in the absence of light, in both linear and logarithmic scales. The illustration is a photograph of a device. (**c**) I–V curves of the device under different wavelengths of light. (**d**) The enlarged I–V curves in the dark and under light of 375 nm. (**e**) Time-resolved optical response characteristics of the device when the bias voltage is equal to 0V for 375 nm illumination. (**f**) The response spectrum of the WS_2_/GaN heterojunction device and the absorption spectra of GaN and WS_2_/GaN. Reproduced with permission from Ref. [116]. Copyright 2019, Royal Society of Chemistry.

**Figure 16 molecules-29-03341-f016:**
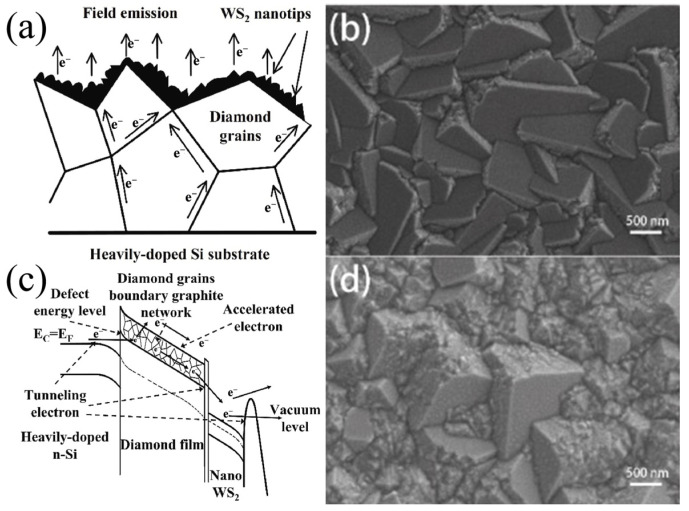
(**a**) Electron transfer diagram of diamond/WS_2_ composite film FE device. (**b**) SEM image of diamond. (**c**) Energy band diagram of WS_2_/ diamond composite film. (**d**) SEM image of WS_2_/diamond composite film. Reproduced with permission from Ref. [14]. Copyright 2024, Elsevier.

**Figure 17 molecules-29-03341-f017:**
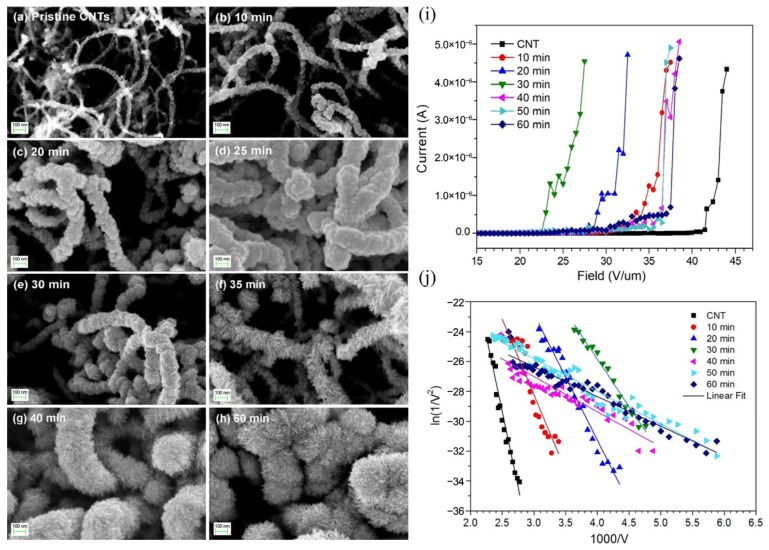
(**a**) Top-view SEM image of the original CNT. (**b**) WS_2_-CNT samples were fabricated with sputter deposition times of 10 min, (**c**) 20 min, (**d**) 25 min, (**e**) 30 min, (**f**) 35 min, (**g**) 40 min, and (**h**) 60 min. (**i**) Current vs. voltage field emission plots. (**j**) Fowler–Nordheim plot of WS_2_-CNT samples. Reproduced with permission from Ref. [126]. Copyright 2019, Springer Nature.

**Figure 18 molecules-29-03341-f018:**
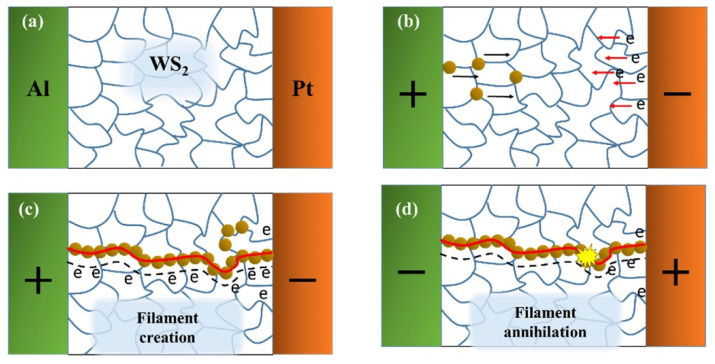
(**a**) Device diagram. (**b**) Charge flowing along grain boundaries. (**c**) Formation of conductive paths (red). (**d**) Breakage of conductive paths. Reproduced with permission from Ref. [129]. Copyright 2020, Elsevier.

**Table 1 molecules-29-03341-t001:** LEDs application of 2D WS_2_ with different morphologies.

Materials	WS_2_ Morphology	Synthesis Route	Application	Ref.
Ion-gel/Au/Ni/WS_2_/PEN	Monolayer	CVD	Polarized LED	[37]
ITO/PEDOT:PSS/poly-TPD/WS_2_/ZnO/Al	Monolayer	MOCVD	OLED	[38]
Au/WS_2_/Si/SiO_2_	Monolayer	Mechanical exfoliation	ACLED	[40]
Ion-gel/Au/WS_2(1−x)_Se_2x_	Monolayer	CVD	Wavelength adjustable LED	[41]

**Table 2 molecules-29-03341-t002:** Sensors application of 2D WS_2_ with different morphologies.

Materials	WS_2_ Morphology	Synthesis Route	Application	Ref.
WS_2_/Ni-In_2_O_3_	Nanosheets	Hydrothermal method	Gas Sensor	[47]
WS_2_/ZnO	Few layers	Laser lift-off	Gas Sensor	[48]
WS_2_/WO_3_	Nanosheets	In situ oxidation method	Gas Sensor	[49]
WS_2_/SnO_2_	Nanosheets	Hydrothermal method	Gas Sensor	[51]
Pd-WS_2_/Si	Few layers	Sputters	Flexible sensors	[61]
WS_2_/Graphene	Monolayer	CVD	Flexible sensors	[62]
RGO/WS_2_	Few layers	Hydrothermal method	Humidity sensor	[63]

**Table 3 molecules-29-03341-t003:** Transistor application of 2D WS_2_ with different morphologies.

Materials	WS_2_ Morphology	Synthesis Route	Application	Ref.
WS_2_	Nanosheets	Mechanical exfoliation	Dual door control FETs	[82]
WS_2_	Bilayer	CVD	MOSFET	[83]
WS_2_/Al_2_O_3_	Nanosheets	Mechanical exfoliation	MIS contact FET	[88]
WS_2_	Monolayer	VLS	P-type channel FETs	[91]
HZO/WS_2_	Nanosheets	Mechanical exfoliation	Synaptic transistors	[95]
WS_2_/MoS_2_	Few layers	CVD and mechanical exfoliation	Phototransistors	[96]

**Table 4 molecules-29-03341-t004:** Photodetector application of 2D WS_2_ with different morphologies.

Materials	WS_2_ Morphology	Synthesis Route	Application	Ref.
TFSA-GR/WS_2_	Few layers	CVD	Flexible photodetectors	[105]
WS_2_	Few layers	RF magnetron sputtering and electron-beam irradiation	Flexible photodetectors	[106]
WS_2_/GaAs	Few layers	Two-step thermal decomposition method	Wide spectrum photodetectors	[108]
WS_2_/PbS	Nanosheets	Mechanical exfoliation	Infrared photodetectors	[115]
RGO/WS_2_	Nanosheets	Hydrothermal reaction	Ultraviolet photodetectors	[117]

**Table 5 molecules-29-03341-t005:** Field emission devices application of 2D WS_2_ with different morphologies.

Materials	WS_2_ Morphology	Synthesis Route	Application	Ref.
WS_2_	Few layers	EBVD	Field emission devices	[14]
WS_2_/Si	Nanotubes	High-temperature solid-gas synthetic	Field emission devices	[124]
WS_2_/CNTs	Few layers	RF Magnetron Sputtering	Field emission devices	[126]

**Table 6 molecules-29-03341-t006:** Non-volatile memory devices application of 2D WS_2_ with different morphologies.

Materials	WS_2_ Morphology	Synthesis Route	Application	Ref.
WS_2_/hBN/Graphene	Nanosheets	Mechanical exfoliation	Non-volatile memory devices	[128]
Al/WS_2_/Pt/Ti	Few layers	CVD	Non-volatile memory devices	[129]
WS_2_/WSe_2_	Monolayer	Mechanical exfoliation	Non-volatile memory devices	[130]
MED WS_2_	Monolayer	Monolayer	Non-volatile memory devices	[131]
ITO/Al_2_O_3_/Ta_2_O_5_/Al_2_O_3_/WS_2_	Few layers	Mechanical exfoliation	Non-volatile memory devices	[132]

## Data Availability

No new data were created.

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
