# Peer review of "Recent Excellent Optoelectronic Applications Based on Two-Dimensional WS_2_ Nanomaterials: A Review"

_molecules, 2024, doi:10.3390/molecules29143341_

Round 1
Reviewer 1 Report
Comments and Suggestions for Authors
Comments:
- WS2 is N-type or p-type?
- What is the effect of N-type or p-type in optoelectronic applications?
- The disadvantages of WS2 and solutions for their overcome should be discussed in a new section.
- Synthesis methods of WS2 should be discussed.
- The effect of morphology on optoelectronic properties should be discussed.
Comments on the Quality of English LanguageModerate editing of English language required
Reviewer 2 Report
Comments and Suggestions for Authors
Reviewer Comments:
The abstract must be improved, and more detail about the narration of the article must be added.
Why did the authors mention "latest 5 years" and cite 2016 refs?
It's crucial to include a section on the properties of WS2 before discussing its application. This will provide a solid foundation for the readers to understand the context and significance of the subsequent application.
Authors must include some of their own schemes for better understanding.
There is no table provided to compare the activity of WS2. When the authors describe sensors and other applications, they must provide tables with the recent reports on WS2 regarding the application.
Some of the keywords are not suitable.
The Intoduction is entirely unacceptable. No information about the material has been provided, and none of the recent achievements of the materials have been discussed.
Also, include a section about materials similar to WS2 for optoelectronic applications.
Comments on the Quality of English Language
Must be improved
Author Response
请参阅附件。

Round 2
Reviewer 1 Report
Comments and Suggestions for Authors
Accept in present form
Comments on the Quality of English LanguageMinor editing of English language required
Reviewer 2 Report
Comments and Suggestions for Authors
The authors have addressed all my comments well. This may be accepted after the editor's review.